

# Influence of basement rocks on fluid evolution during multiphase deformation: the example of the Estamariu thrust in the Pyrenean Axial Zone

Daniel Muñoz-López[1], Gemma Alías[1], David Cruset[2], Irene Cantarero[1], Cédric M. Jonh[3], Anna Travé[1]

[1]Department de Mineralogia, Petrologia i Geologia Aplicada. Facultat de Ciències de la Terra, Universitat de Barcelona (UB), C/ Martí i Franquès s/n, 08028 Barcelona, Spain.
[2]Institut de Ciències de la Terra Jaume Almera, ICTJA-CSIC, Lluís Solé i Sabarís s/n, 08028 Barcelona, Spain.
[3] Department of Earth Science and Engineering, Imperial College London, London SW7 2BP, UK.

*Correspondence to*: Daniel Muñoz-López (munoz-lopez@ub.edu)

**Abstract.** Calcite veins precipitated in the Estamariu thrust during two tectonic events decipher the temporal and spatial relationships between deformation and fluid migration in a long-lived thrust and determine the influence of basement rocks on the fluid chemistry during deformation. Structural and petrological observations constrain the timing of fluid migration and vein formation, whilst geochemical analyses ($\delta^{13}C$, $\delta^{18}O$, $^{87}Sr/^{86}Sr$, clumped isotope thermometry and elemental composition)

of the related calcite cements and host rocks indicate the fluid origin, pathways and extent of fluid-rock interaction. The first tectonic event, recorded by calcite cements Cc1a and Cc2, is related to the Alpine reactivation of the Estamariu thrust, and is characterized by the migration of meteoric fluids, heated at depth (temperatures between 56 and 98 ºC) and interacted with crystalline basement rocks before upflowing through the thrust zone. During the Neogene extension, the Estamariu thrust was reactivated and normal faults and shear fractures with calcite cements Cc3, Cc4 and Cc5 developed. Cc3 and Cc4 precipitated

from hydrothermal fluids (temperatures between 127 and 208 ºC and between 102 and 167 ºC, respectively) derived from crystalline basement rocks and expelled through fault zones during deformation. Cc5 precipitated from low temperature meteoric waters percolating from the surface through small shear fractures. The comparison between our results and already published data in other structures from the Pyrenees suggests that regardless of the origin of the fluids and the tectonic context, basement rocks have a significant influence on the fluid chemistry, particularly on the $^{87}Sr/^{86}Sr$ ratio. Accordingly, the cements

precipitated from fluids interacted with crystalline basement rocks have significantly higher $^{87}Sr/^{86}Sr$ ratios ($> 0.710$) with respect to those precipitated from fluids that have interacted with the sedimentary cover ($< 0.710$).

## 1 Introduction

Deformation associated with crustal shortening is mainly accommodated by thrust faulting and related fault zone structures. Successive faulting may occur and favourably oriented structures may undergo reactivation during different tectonic events in

a long-lived orogenic belt (Cochelin et al., 2018; Sibson, 1995). The reactivation of faults may produce changes in the hydraulic



behaviour of fault zones as well as in the origin and regime of fluids circulating through them. Consequently, constraining the timing of deformation and fluid migration is essential to better understand the current configuration of a mountain belt, its evolution through time and the mobilization of different fluids during successive deformation events (Baqués et al., 2012; Crespo-Blanc et al., 1995; Faÿ-Gomord et al., 2018; Fitz-Diaz et al., 2011; Lacroix et al., 2014). Understanding basin-scale

fluid flow is of primary importance to reconstruct the diagenetic history of a sedimentary basin, as fluids take part in a wide range of geological processes including precipitation of new mineral phases, dolomitization and petroleum migration, among others (Barker et al., 2009; Foden, 2001; Fontana et al., 2014; Gomez-Rivas et al., 2014; Martín-Martín et al., 2015; Mozafari et al., 2019; Piessens et al., 2002). Due to the economic interest of these processes, in particular related to oil and ore deposits exploration, $CO_2$ sequestration, seismic activity and water management, many researchers have addressed the study of the

relationship between deformation and fluid migration (Beaudoin et al., 2014; Breesch et al., 2009; Cox, 2007; Dewever et al., 2013; Gasparrini et al., 2013; Suchy et al., 2000; Travé et al., 2009; Voicu et al., 2000; Warren et al., 2014).

In the Pyrenees, the crystalline basement rocks from the Axial Zone are affected by numerous fault systems considered Variscan in age but reactivated during the Pyrenean compression (Cochelin et al., 2018; Poblet, 1991). However, no real consensus exists about the influence of the Alpine deformation on the basement rocks and the age of basement-involved

structures is still debated (Cochelin et al., 2018; García-Sansegundo et al., 2011). As a consequence, the relationships between deformation and fluid flow have been focused on structures from the Mesozoic and Cenozoic cover (Crognier et al., 2018; Cruset et al., 2016, 2018; Beaudoin et al., 2015; Lacroix et al., 2011, 2014; Nardini et al., 2019; Travé et al., 1997, 1998, 2000, 2007), where the timing of deformation and thrust emplacement is well-constrained (Vergés et al., 2002; Vergés, 1993; Vergés and Muñoz, 1990). Contrarily, only a few contributions, concentrated along the Gavarnie thrust system, have tackled this topic

in the Paleozoic basement involving crystalline rocks (Banks et al., 1991; Grant et al., 1990; Henderson and McCaig, 1996; McCaig et al., 2000a, 1995; Rye and Bradbury, 1988; Trincal et al., 2017). On the other hand, another important aspect of studying the fault-fluid system is related to the heat flow and the influence of faults on the development of geothermal systems (Faulds et al., 2010; Grasby and Hutcheon, 2001; Liotta et al., 2010; Rowland and Sibson, 2004). Particularly, in the NE part of the Iberian Peninsula (including the Pyrenees and the Catalan Coastal Ranges), the presence of high-permeability Neogene

extensional faults, acting as conduits for focused upward migration, provide efficient pathways for hydrothermal fluids to flow from deeper to shallower crustal levels (Carmona et al., 2000; Fernàndez and Banda, 1990; Taillefer et al., 2017, 2018). In this sense, understanding the fault-fluid system evolution and the timing of hydrothermal fluid migration is of great importance to characterize the potential geothermal resources of this area.

In this contribution, we report the temporal and spatial relationships between deformation and fluid migration in a long-lived

Variscan thrust deforming basement rocks in the Pyrenean Axial Zone. For this purpose, we combine structural, petrological and geochemical analyses of calcite veins precipitated in the Estamariu thrust during two episodes of regional tectonic activity (Pyrenean compression and Neogene extension). Structural and petrological observations allow us to unravel the timing of fluid migration and vein formation in relation to the involved tectonic events. The geochemistry of the vein cements and related host rocks provides information on the fluid origin, pathways and extent of fluid-rock interaction during deformation. The



integration of all these data within the studied geological setting in the Pyrenean basement will allow us: (i) to constrain the timing of vein formation and fluid migration; (ii) to determine the fluid origin and pathways during successive compressional and extensional deformation phases; (iii) to assess the influence of basement rocks on the chemistry of fluids circulating during deformation; and (iv) to provide insights into the fluid flow at regional scale in the NE part of the Iberian Peninsula, where the presence of hydrothermal fluids has been reported from Neogene times to present.

## 2 Geological setting

The Pyrenees constitute an asymmetric and doubly verging orogenic belt developed from the Late Cretaceous to Oligocene in relation to the Alpine convergence between the Iberian and European plates (Choukroune, 1989; Muñoz, 1992; Roure et al., 1989; Sibuet et al., 2004; Srivastava et al., 1990; Vergés and Fernàndez, 2012). The Pyrenean structure consists of a central antiformal stack of basement-involved rocks from the Axial Zone, flanked by two oppositely verging fold-and-thrust belts and
their associated foreland basins (Muñoz, 1992; Muñoz et al., 1986) (Fig. 1A). The Axial Zone represents a fragment of the European Variscan orogeny incorporated into the Pyrenean belt during the Alpine convergence (Matte, 1991). It consists of a duplex of three south-directed basement thrust sheets (from upper to lower: Nogueres, Orri and Rialp), involving Cambrian to Carboniferous rocks deformed by the Variscan compressional events, and an Upper Carboniferous to Triassic post-Variscan cover (Poblet, 1991; Saura, 2004; Saura and Teixell, 2006).
During the Neogene tectonic evolution of the eastern Axial Pyrenees, a horst and basin system bounded by E-W to ENE-WSW faults developed (Roca, 1996; Roca and Guimerà, 1992). The most important fault, La Tet fault (roughly striking N60°E and dipping around 60°N) has associated a set of E-W extensional basins such as La Cerdanya, Conflent, La Seu d'Urgell and Cerc basins (Cabrera et al., 1988; Roca, 1996). The Cerc basin consists of a Stephano-Permian accumulation of volcanic rocks discordantly overlying Cambro-Ordovician materials from the Orri thrust sheet. This basin is thrusted in its eastern limit by
the Estamariu thrust, whereas the northern and southern boundaries correspond to two Neogene extensional faults, La Seu d'Urgell fault and the Ortedó fault, respectively (Hartevelt, 1970; Roca, 1996; Saura, 2004) (Fig. 1B, C). In the NW part of the basin, the limit between the Stephano-Permian unit and the Upper Ordovician sequence corresponds to a Stephano-Permian extensional fault formed coevally with the deposition of the volcanic sequence (Saura, 2004). This fault was reactivated during the latest stages of the Neogene extension (Saura, 2004) and is here referred as the Sant Antoni fault (Fig. 1C).
The Estamariu thrust is a basement-involved reverse fault originated during the Variscan orogeny with a minimum displacement of 27 km (Poblet, 1991). However, in its southwestern termination, it juxtaposes the Devonian Rueda Formation against the Stephano-Permian Erill Castell Formation. The Erill Castell Formation developed during the late to post-orogenic collapse of the Variscan belt (Lago et al., 2004; Ziegler, 1988), evidencing the reactivation of the Estamariu thrust during the Alpine Orogeny (Poblet, 1991; Saura, 2004).
Rocks cropping out around the Estamariu thrust and the Cerc basin range from Upper Ordovician to Miocene (Fig. 1C) and are deformed by successive Variscan, Alpine and Neogene phases (Saura and Teixell, 2006). Due to such a complex structural



setting, the stratigraphic record is discontinuous and only Upper Ordovician, Devonian, Stephano-Permian and Neogene successions are present in the study area. The Upper Ordovician consists mainly of a detrital sedimentary sequence affected by a low-graded metamorphism (Hartevelt, 1970; Poblet, 1991; Saura, 2004). This succession basically includes a centimetric
to metric alternation of shales, sandstones, conglomerates, quartzites and phyllites. The Devonian succession consists of a centimetric to decimetric alternation of limestones and black slates (Rueda Formation) (Mey, 1967). The Stephano-Permian sequence is represented by a volcanic and volcanoclastic unit (the Erill Castell Formation) (Mey et al., 1968), involving tuffs and ignimbrites at the base and andesites in the upper part (Martí, 1996; Saura and Teixell, 2006). Finally, the Neogene sequence is constituted of detrital and poorly lithified sediments, mainly shales, sandstones and conglomerates (Roca, 1996).

## 3 Methods

This study integrates a field compilation of structural data and samples and petrological and geochemical analyses of calcite cements and related host rocks. The structural data includes bedding, foliation strike and fracture orientation, type, crosscutting relationships and kinematics. Such data were plotted in equal-area lower-hemisphere projections and integrated in a schematic map and a cross-section of the Estamariu thrust and the Cerc basin (Fig. 2A, B and 3). Host rocks and calcite cements were
sampled for petrological and geochemical analyses. Thin sections were studied under a Zeiss Axiophot optical microscope and a Cold cathodoluminescence (CL) microscope model 8200 Mk5-1 operating between 16–19 kV and 350 µA gun current. Thirty-five representative samples of the different generations of calcite cements and the carbonate portion of the Devonian rocks were sampled for carbon and oxygen isotopy. Sampling was carried out with a 500 µm-diameter microdrill. 50-100 µg of powdered samples were reacted with 100% phosphoric acid during two minutes at 70 ºC. The resultant $CO_2$ was analyzed
with an automated Kiel Carbonate Device attached to a Thermal Ionization Mass Spectrometer Thermo Electron MAT-252 (Thermo Fisher Scientific) according to the method of McCrea (1950). For calibration, the International Standard NBS-18 and the internal standard RC-1, traceable to the International Standard NBS-19, were used. The standard deviation is ±0.03‰ for $\delta^{13}C$ and ±0.05‰ for $\delta^{18}O$ expressed with respect to the VPDB standard (Vienna Pee Dee Belemnite).

The elemental composition of twelve samples of calcite cements and related host rocks were analyzed using a high resolution
inductively coupled plasma-mass spectrometry (HR-ICP-MS, model Element XR, Thermo Fisher Scientific). 100 mg of powdered samples were dried at 40 ºC during 24 h and then, they were acid digested in closed polytetrafluoroethylene (PTFE) vessels with a combination of $HNO_3 + HF + HClO_4$ (2.5 mL: 5 mL: 2.5 mL v/v). The samples were evaporated and 1 mL of $HNO_3$ was added to make a double evaporation. Finally, the samples were re-dissolved and diluted with MilliQ water (18.2 MΩ cm$^{-1}$) and 1 mL of $HNO_3$ in a 100 mL volume flask. A tuning solution containing 1 g L$^{-1}$ Li, B, Na, K, Sc, Fe, Co, Cu,
Ga, Y, Rh, In, Ba, Tl, U was used in order to improve the sensitivity of the ICP-MS , and as internal standard, 20 mg L$^{-1}$ of a monoelemental solution of $^{115}$In. Reference materials are the BCS-CRM nº 393 (ECRM 752-1) limestone, JA-2 Andesite and JB-3 Basalt. The precision of the results was expressed in terms of two standard deviations of a set of eight reference materials measurements (reference material JA-2), whereas accuracy (%) was calculated using the absolute value of the difference





between the measured values obtained during the analysis and the certified values of a set of eight reference material analysis
(reference material BCS-CRM nº 393 for major oxides and JA-2 for trace elements). The detection limit (DL) was calculated
as three times the standard deviation of the average of ten blanks.

The $^{87}$Sr/$^{86}$Sr ratio was analyzed for eight representative samples of calcite cements and host rocks. Powdered samples were
dissolved in 5 mL of 10% acetic acid. After centrifugation, the supernatant was dried and dissolved in 1 mL of 1M HNO$_3$. The
solid residue generated after evaporation was diluted in 3 mL of 3M HNO$_3$ and loaded into chromatographic columns to
separate the Rb-free Sr fraction, using SrResinTM (crown-ether (4,4'(5')-di-t-butylcyclohexano-18-crown-6)) and 0.05M
HNO$_3$ as eluent. After evaporation, samples were loaded onto a Re filament along with 1 μL of 1 M phosphoric acid and 2 μL
of Ta$_2$O$_5$. Isotopic analyses were carried out in a TIMS-Phoenix mass spectrometer performing a dynamic multicollection
method, during 10 blocks of 16 cycles each one, keeping a $^{88}$Sr beam intensity of 3-V. Possible $^{87}$Rb interferences and possible
mass fractionation during sample loading and analysis were corrected and normalized with the reference value of $^{88}$Sr/$^{86}$Sr =
0.1194. The isotopic standard NBS-987 was analyzed six times during sample analysis, yielding an average value of 0.710243
±0.000009 (standard deviation, 2σ). NBS 987 data have been used to correct the sample ratios for standard drift from the
certified value. The analytical error in the $^{87}$Sr/$^{86}$Sr ratio, referred to two standard deviations, was 0.01%, whereas the internal
precision is 0.000003. Sr procedural blanks were always below 0.5 ng.

The $^{143}$Nd/$^{144}$Nd ratios were analyzed in seven samples of calcite cements and host rocks.  Samples were weighted in Teflon®
vessels, with enriched spike solution ($^{149}$Sm-$^{150}$Nd - Oak Ridge) and dissolved in 5 mL of ultrapure HF and 3 mL of ultrapure
HNO$_3$ (Merck-Suprapur$^{TM}$). The PFA-vessels were placed 65 hours at 120 ºC into an oven. After that, cold vials were
evaporated at 120 ºC on a heat plate. 4 mL of 6N distilled HCL were added to the dried samples and placed at 120 ºC in an
oven overnight. The solid residue generated after evaporation was dissolved in 3 mL of distilled and titrated 2.5 N HCl.
Samples were centrifuged at 4000 rpm for 10 minutes to separate the possible dissolved fraction from the residue.
Chromatographic separation of the whole group of REE was performed with a previously calibrated cation exchange resin
DOWEX 50W-X8 200-400 mesh. After that, recovered REE fractions were dried and again dissolved in 200 μL HCl 0.18N.
Such solutions were passed in a new chromatographic step (Ln-resin). The result is a complete separation between the Nd and
the Sm fractions, using 0.3N HCl and 0.4N HCl as eluent, respectively. Dried Sm and Nd samples dissolved with 2 μL of
0.05M phosphoric acid were loaded onto a side Rhenium (Re) filament of a triple Re filament arrangement. Nd ratios were
analysed in a mass spectrometer TIMS-Phoenix®, using a dynamic multicollection method, through 160 cycles at a stable
intensity of 1V for the $^{144}$Nd mass. In turn, Sm ratios were analysed in the same spectrometer, using a single static method
through 112 cycles keeping 1V intensity for the $^{149}$Sm mass. Nd analyses were corrected for $^{142}$Ce and $^{144}$Sm interferences, if
any, and normalized to a ratio of $^{146}$Nd/$^{144}$Nd = 0.7219 to correct the possible mass fractionation during the processes of loading
and analysing at the TIMS. Nd isotopic standard JNdi-1 was checked to correct the sample ratios for standard drift from the
certified value. The analytical error (2STD) was 0.1% in the $^{147}$Sm/$^{144}$Nd ratio and 0.006% in the $^{143}$Nd/$^{144}$Nd ratio. Procedural
blanks were always below 0.1 ng.





U-Pb geochronology of the calcite cements Cc1a to Cc5 was accomplished at FIERCE (Frankfurt Isotope and Element Research Center, Goethe University) using a laser ablation inductively coupled plasma mass spectrometry (LA-ICPMS). Clumped isotope thermometry of the calcite cements was carried out in order to determine the temperature and composition

($\delta^{18}O_{fluid}$) of the vein-forming fluids. 2–3 mg aliquots from cements were measured with an automated line, the Imperial Batch Extraction system (IBEX), developed at Imperial College. Samples were dropped in 105% phosphoric acid at 90 ºC and reacted during 30 min. The reactant $CO_2$ was separated with a poropak-Q column and transferred into the bellows of a Thermo Scientific MAT 253 mass spectrometer. The characterization of a replicate consisted of 8 acquisitions in dual inlet mode with 7 cycles per acquisition. The post-acquisition processing was completed with Easotope, a software for clumped isotope

analyses (John and Bowen, 2016). During phosphoric acid digestion, $\Delta_{47}$ values were corrected for isotope fractionation using a phosphoric acid correction of 0.069‰ at 90 ºC for calcite (Guo et al., 2009). The data were also corrected for non-linearity applying the heated gas method (Huntington et al., 2009) and projected into the reference frame of Dennis et al., (2011). Carbonate $\delta^{18}O$ values were calculated with the acid fractionation factors of Kim and O'Neil, (1997). Results were converted to temperatures applying the calibration method of Kluge et al., (2015). Calculated $\delta^{18}O_{fluid}$ values are expressed in ‰ with

respect to the Vienna Standard Mean Ocean Water (VSMOW).

## 4 Results

### 4.1 Structure and associated calcite cements

The Estamariu thrust strikes N-S to NW-SE and dips between 40 and 70º towards the NE. It has a displacement of a few hundred meters and juxtaposes a Devonian alternation of limestones and shales in the hanging wall against Stephano-Permian

andesites in the footwall (Poblet, 1991) (Fig. 2-4). The main slip plane is undulose, producing changes in the strike direction and dip, and generates a 2 – 3 m thick thrust zone, which is thicker in the hanging wall, up to 2.5 m thick. In the footwall, the thrust zone is less than 1 m thick and has associated minor restricted thrust zones developed as subsidiary accommodation structures related to the main thrust fault (Fig. 2A, B). All kinematic indicators, including S-C structures and slickenlines, indicate reverse displacement towards the west.

The mesostructures and microstructures observed in the study area are described below according to their structural position in relation to the Estamariu thrust, that is, hanging wall, thrust zone and footwall (Fig. 3, 4).

In the studied outcrops, the Devonian Rueda Formation from the hanging wall is characterized by a well-bedded alternation of dark to light grey limestones with dark grey shales (Fig. 5A). Limestones are more abundant and are made up of encrinites, which consist of a bioclastic packstone formed essentially of crinoid stems (Fig. 5B). Under cathodoluminescence, encrinites

show dark to bright orange colors (Fig. 5C). Deformation in the Devonian protolith (i.e., outside the thrust zone) corresponds to a decametric anticline (Fig. 2B), which has associated an axial plane pervasive regional foliation (Sr) concentrated in the pelitic intervals (Fig. 5B). The Sr has a NW-SE mean orientation, dips 30 to 55º towards the E and NE and is generally between



2 and 5 cm spaced. The angular relationship between the Devonian bedding strike and the regional foliation is oblique and changes to nearly perpendicular when approaching the thrust zone (Fig. 2B).

Within the thrust zone affecting the hanging wall, the Devonian host rocks are still recognizable, but deformation intensity progressively increases towards the main thrust plane. This deformation consists of a penetrative thrust zone foliation ($S_D$), two generations of stylolites (e1, e2) and three generations of calcite veins (V0, V1a and V1b) (Fig. 3, 6). These structures are described below in chronological order.

The foliation within the thrust zone affecting the Devonian hanging wall strikes NW-SE and dips $40 – 50°$ NE, similar to the
regional foliation in the protolith, but it is more closely spaced, generally between 0.2 and 1 cm (Fig. 6A, B). This observation points out to a progressive transposition of the regional foliation within the thrust zone during thrusting. At mesoscale, $S_D$ has related shear surfaces (Ci) defining centimetric S-C-type structures, again indicating reverse kinematics (Fig. 6A).

Stylolites e1 have a wave-like shape and trend subparallel to the thrust zone foliation (Fig. 6B, C). When present, these stylolites are very systematic with densities between 5 and 8 stylolites/cm.

The first calcite vein generation (V0), only observed at microscopic scale (Fig. 6B, C), corresponds to up to 1 cm long and less than 1 mm thick veins cemented by blocky to elongate blocky calcite crystals featuring a dark-brown luminescence (cement Cc0). Veins V0 and stylolites e1 are perpendicular between them and show ambiguous crosscutting relationships. These microstructures are concentrated into discontinuous fragments of the Devonian host rocks within the thrust zone. Calcite veins V1a crosscut the previous vein generation (V0) and are developed within $S_D$ surfaces (Fig. 3, 6D). These veins are the most
abundant, exhibit a white to brownish color in hand sample and are up to 10 cm long and 1 cm thick. The vein cement (Cc1a) is formed of up to 3-4 mm in size anhedral crystals displaying a blocky texture and a dark brown luminescence (Fig. 6E).

Stylolites e2, more abundant than stylolites e1, are up to 10 cm long and show spacing between 0.5 and 2 cm (Fig. 6D, F). These stylolites mainly correspond to sutured areas developed between the host rock and the calcite veins V1a and between foliation surfaces.

Calcite veins V1b, up to 1 cm long and less than one mm thick, were also identified at microscopic scale (Fig. 6D, F). The vein cement (Cc1b) consists of up to 0.1 mm calcite crystals with a blocky texture and a bright yellow luminescence. These veins postdate the previous V0 and V1a generations and trend perpendicular to stylolites e2.

Towards the fault plane, the thrust zone foliation is progressively more closely-spaced and stylolites e2 become more abundant (showing mm spacing) and exhibit ambiguous cross-cutting relationships with veins V1b (Fig. 6F). The main slip surface
corresponds to a discrete plane that contains calcite slickensides (veins V2). The vein cement (Cc2) is milky white in hand sample and consists of up to 3 mm blocky to elongated blocky crystals (Fig. 6G) with a dull to bright orange luminescence (Fig. 6H).

Deformation in the footwall is concentrated within the main thrust zone and subsidiary thrust zones and corresponds to a thrust zone foliation ($S_{SP}$) and calcite veins V3 (Fig. 3).

The foliation strikes NW-SE, dips towards the NE and is mm to cm spaced (Fig. 7A). Calcite veins V3 are generally $1 – 2$ cm thick and strike NW-SE. They are parallel or locally branch off cutting the foliation planes in the subsidiary thrust zone (Fig.





7A, B). Outside the thrust zone, veins V3 are locally present but have a NE-SW strike. These veins are mostly less than 1 m long and are spaced between a few cm and 50 cm. The vein cement (Cc3) is made up of a milky white calcite characterized by up to 3 mm long fibrous crystals oriented perpendicular to the vein walls (Fig. 7C). Locally, anhedral blocky crystals

ranging in size from 0.1 to 1 mm are also present. This cement displays a bright yellow to bright orange luminescence (Fig. 7D).

In the footwall, the Stephano-Permian Erill Castell Formation comprises massive, dark-greenish andesitic levels showing a rhythmic magmatic layering (Sm) (Fig. 7E), which corresponds to a fluidal structure of the host rock. The local presence of pyroclastic and brecciated volcanoclastic levels is also ubiquitous mainly in the lower part of this sequence. Andesites are

characterized by a porphyritic texture defined by a dark fine-grained spherulitic matrix partially devitrified and large zoned crystals of plagioclase (Fig. 7F), up to 2 – 3 cm long, and less abundant biotite and hornblende. These mafic phenocrysts are systematically pseudomorphosed by clay minerals and frequently show evidence of oxidation and chloritization. Andesites are affected by E-W striking open joints (J1) dipping indistinctively towards the north and south (Fig. 7E). These joints locally trend parallel to the magmatic layering (Fig. 3).

Finally, as described above, the northern and southern limits of both the Cerc basin and the Estamariu thrust correspond to two Neogene extensional faults, La Seu d'Urgell and the Ortedó fault systems (Fig 1C). These faults are subvertical or greatly dip towards the north. In the northern part, the slip plane of La Seu d'Urgell fault has not been observed and the limit between the Stephano-Permian rocks and the Neogene deposits is not well constrained due to the poor quality of the Neogene outcrops and the presence of Quaternary deposits. In the southern part, the Ortedó fault generates a several meter-thick dark greyish to

brown fault zone, characterized by the presence of clay-rich incohesive fault rocks developed at the contact between Stephano-Permian and Upper Ordovician rocks. Related to these main fault systems, mesoscale normal faults (V4) are also frequent affecting the andesites within the Cerc basin. These faults are mainly E-W and locally NE-SW, are subvertical and dip indistinctly towards the N and S. Fault planes are locally mineralized with calcite cement and exhibit two striae set generations indicating dip-slip and strike-slip movements (Fig. 8A). The calcite cement (Cc4) consists of up to 2 mm blocky to elongated

blocky crystals (Fig. 8B) with a homogeneous dark orange luminescence (Fig. 8C). On the other hand, the main Estamariu thrust zone is locally displaced by shear fractures (V5) (Fig. 8D) and a later set of shear bands (Cn) (Fig. 8E), both having an overall NNW-SSE to NNE-SSW strike (Fig. 3) that indicate a minor normal displacement. Shear fractures (V5) are locally mineralized with a greyish microsparite calcite cement (Cc5).

## 4.2 Geochemistry of calcite cements and host rocks

The geochemistry ($\delta^{18}O$, $\delta^{13}C$, $\delta^{18}O_{fluid}$, $^{87}Sr/^{86}Sr$, $^{143}Nd/^{144}Nd$ and elemental composition) and the calculated temperature of precipitation of the different calcite cements Cc1a to Cc5 are described below. Veins V0 and V1b were only observed at microscopic scale and their calcite cement Cc0 and Cc1b could not be sampled to perform these geochemical analyses.

The $\delta^{18}O$ and $\delta^{13}C$ isotopic composition of the carbonate fraction of the Devonian hanging wall and the different calcite cements (Cc1a to Cc5) are summarized in Table 1 and represented in Fig. 9. The micritic matrix of the Devonian packstone





ranges in $\delta^{18}O$ values between -10.5 and -8.4 ‰VPDB and in $\delta^{13}C$ values between +1.5 and +2.8 ‰VPDB, whereas the calcite

cements have a broader range of values depending on the cement generation (Fig. 9).

Calcite cement Cc1a has $\delta^{18}O$ values between -11.3 and -10.3 ‰VPDB and $\delta^{13}C$ values between +0.8 and +2.1 ‰VPDB. Cc2

is characterized by $\delta^{18}O$ values between -14.9 and -12.9 ‰VPDB and $\delta^{13}C$ values between -1.2 and +1.5 ‰VPDB. Cc3 has

$\delta^{18}O$ values between -14.3 and -13.4 ‰VPDB and $\delta^{13}C$ values between -9.3 and -6.9 ‰VPDB. Cc4 exhibits $\delta^{18}O$ values

between -13.8 and -13.4 ‰VPDB and $\delta^{13}C$ values between -7.4 and -7.2 ‰VPDB and Cc5 ranges in $\delta^{18}O$ between -8.1 and -

5.7 ‰VPDB and in $\delta^{13}C$ between -8.2 and -3.8 ‰VPDB. The calcite cement Cc1a, precipitated in the fault zone affecting the

Devonian hanging wall, has enriched $\delta^{13}C$ values, whilst the calcite cement within the fault plane (Cc2) exhibits either negative

or positive $\delta^{13}C$ values and the calcite cements hosted in the Stephano-Permian andesites (Cc3 to Cc5) have more depleted

$\delta^{13}C$ values (Fig. 9). In addition, calcite cements show a progressive depletion in $\delta^{18}O$ from Cc1a to Cc4, whereas Cc5 displays

more enriched $\delta^{18}O$ values.

The obtained $\Delta_{47}$ values from clumped isotope thermometry were converted into temperatures and $\delta^{18}O_{fluid}$ (Table 1 and Fig.

10) using the equation of Kluge et al., (2015) and Friedman and O'Neil, (1977), respectively. Cc2 has a $\Delta_{47}$ of 0.567, which

translates into a temperature between 56 and 98 ℃ and a $\delta^{18}O_{fluid}$ between -6.4 and -0.3 ‰VSMOW. For Cc3, $\Delta_{47}$ is 0.045 and

the calculated T and $\delta^{18}O_{fluid}$ are 127 to 208 ℃ and +4.3 to +12.1 ‰VSMOW, respectively. $\Delta_{47}$ for Cc4 is 0.48, which translates

into a T and a $\delta^{18}O_{fluid}$ between 102 to 167 ℃ and +0.9 to +8.1 ‰VSMOW, respectively. For Cc5 the $\Delta_{47}$ is 0.77 and the

calculated T and a $\delta^{18}O_{fluid}$ are between -5 and +3 ℃ and between -12.4 and -10.1 ‰VSMOW, respectively.

The $^{87}Sr/^{86}Sr$ ratio of calcite cements Cc1a to Cc5 and host rocks are reported in Table 1 and Fig. 11. Devonian limestones

from the hanging wall have a $^{87}Sr/^{86}Sr$ ratio of 0.710663, whilst the Stephano-Permian andesites in the footwall exhibit a more

radiogenic $^{87}Sr/^{86}Sr$ ratio of 0.743983. The calcite cements have more radiogenic $^{87}Sr/^{86}Sr$ ratios than the Devonian limestones

but less radiogenic values than the Stephano-Permian andesites. This ratio ranges from 0.713018 to 0.714092 in Cc1a, is

0.718294 for Cc2, 0.714619 for Cc3, 0.717706 for Cc4 and 0.716923 for Cc5. These results are compared with already

published data from synkinematic veins and deformed rocks in other Pyrenean structures developed in the basement and in the

sedimentary cover during the Pyrenean compression (Fig. 11). This comparison shows that values obtained in this study are:

1) significantly more radiogenic than the values of marine carbonates and synkinematic veins precipitated in the sedimentary

cover (i.e., in the South Pyrenean fault and thrust belt); and, 2) within the same range of values of synkinematic veins and

deformed rocks in the Pyrenean basement (Axial Zone).

The analyzed samples for $^{143}Nd/^{144}Nd$ ratios in calcite cements and host rocks are reported in Table 1. However, due to the

general low Nd concentrations in most of the analyzed calcite cements and the limited amount of powdered samples that were

available, only calcite cement Cc5 and the andesite host rock (footwall) could be measured. Cc5 has a $^{143}Nd/^{144}Nd$ ratio of

0.512178, which is similar to the one of its footwall host rocks, which is 0.512196.

The obtained elemental composition broadly varies among the different calcite cements and related host rocks (Table 2 and

Fig. 12). In the thrust zone affecting the hanging wall, calcite cement Cc1a shows a similar trend to that of the Devonian

limestones, both having high Sr, intermediate-high Mg and Fe and low Mn contents (Fig. 12). In the main thrust plane, calcite



cement Cc2 has low Mg and Fe and intermediate Mn and Sr contents with respect to the other cements. In the footwall, Cc3

and Cc4 have similar elemental composition, characterized by high Mn, intermediate-high Sr, intermediate-low Fe and low

Mg contents. Finally, calcite cement Cc5 follows a similar trend to that of the Stephano-Permian andesites, both having the

highest Fe and Mg and the lowest Sr and Mn contents with respect to the other cements and host rocks.

## 5 Discussion

### 5.1 Chronology of the observed structures

The Estamariu thrust, affecting basement rocks in the Axial Pyrenees, resulted from a long-lived tectonic history that lasted

from Variscan to Neogene times. U-Pb dating of calcite cements Cc1a to Cc5 failed due to their high lead contents and low

uranium contents and therefore, the timing of the different mesostructures and microsturcures has been determined by means

of crosscutting relationships and microstructural analysis.

The Paleozoic metasedimentary rocks from the Pyrenean basement are broadly affected by multiscale folds and axial plane

regional foliation, developed during the main Variscan deformation phase (Bons, 1988; Cochelin et al., 2018; Zwart, 1986).

Similar structures, a decametric-scale anticline and pervasive axial plane foliation, are found in the Devonian sequence located

in the thrust hanging wall (Fig. 2B) and therefore, we consider them to be developed during the Variscan compression,

contemporaneous with the main activity of the Estamariu thrust. Veins V0 are perpendicular to stylolites e1 and show

ambiguous crosscutting relationships between them. Thus, they are interpreted as originated coevally. Both microstructures

are concentrated into discontinuous fragments of the Devonian host rocks and are therefore considered inherited

microstructures likely developed in Variscan times. However, as pointed above, in the study area the Estamariu thrust affects

late- to post-Variscan Stephano-Permian andesites, confirming thus its reactivation during the Alpine orogeny. Accordingly,

the structures that are strictly attributed to the Alpine reactivation of the thrust are those structures indicating reverse kinematics

or associated with a compressional stress, which are found within the thrust zone deformation, at the contact between Devonian

and Stephano-Permian units. Contrarily, the magmatic layering and joints J1 are broadly present in the andesitic footwall,

outside the thrust zone, and in other Stephano-Permian basins, and are therefore considered inherited fluidal and cooling

structures, respectively. For this reason, the calcite veins V1a and V2 (and related cements Cc1a and Cc2), exclusively

occurring within the thrust zone, have been associated with the reactivation of the Estamariu thrust. During this period, and

associated with ongoing deformation and progressive shortening, stylolites e2 developed as sutured areas between host rock

and veins V1a and between foliation surfaces, coevally with the development of veins V1b, as denoted by their crosscutting

relationships and orientations.

Other structures present in the study area, such as veins V3 to V5 and related cements Cc3 to Cc5, are attributed to the Neogene

extension. Veins V3 precipitated in the subsidiary thrust zone developed in the footwall of the Estamariu thrust. These veins

strike parallel to the thrust zone foliation (Fig 7A, B) but are characterized by calcite fibers growing perpendicular to the vein

walls and to the foliation surfaces (Fig. 7C), thus evidencing their extensional character. The presence of extensional calcite



veins opened along previously formed foliation surfaces in a thrust zone has been reported in other structures in the Pyrenees and has been considered to postdate the thrust activity (Lacroix et al., 2011, 2014). Veins V4 precipitated in subvertical and E-W mesoscale faults affecting the Stephano-Permian andesites (Fig. 8), outside the thrust zone (Fig. 3). The fault orientation and dip and the two striae set generations observed on the fault planes are compatible with the Neogene extensional faults that

bound the Cerc basin and postdate the Estamariu thrust (Cabrera et al., 1988; Roca, 1996; Saura, 2004). calcite cements Cc3 and Cc4, occluding veins V3 and V4, have a similar geochemical composition (Fig. 9 – 12), supporting that their precipitation occurred during the same tectonic event and associated with a similar fluid regime. Finally, veins V5 (and related cement Cc5) precipitated locally in shear fractures crosscutting and postdating the thrust-related deformation (Fig. 3, 4 and 8D). These veins strike parallel to the shear bands (Cn) located in the main thrust zone (Fig. 8E), exhibiting normal slip kinematics, postdating

the reverse structures and therefore indicating reactivation of the Estamariu thrust during the Neogene extension.

**5.2 Fluid system during the Alpine reactivation of the Estamariu thrust**

As veins V1a and V2 developed during the Alpine reactivation of the Estamariu thrust, the geochemistry of their related calcite cements Cc1a and Cc2 record the fluid system during this tectonic event.

Calcite cements Cc1a and Cc2 are characterized by high $^{87}Sr/^{86}Sr$ ratios (from 0.713 to 0.714 for Cc1a and 0.718 for Cc2),

significantly more radiogenic than ratios of Phanerozoic seawater (between 0.7070 and 0.7090) (McArthur et al., 2012). This may reflect the incorporation of radiogenic Sr from a fluid derived or interacted with Rb-rich crystalline basement rocks such as those underlying the Estamariu thrust. Previous studies reported similar $^{87}Sr/^{86}Sr$ ratios in crystalline rocks and in synkinematic veins developed in the Pyrenean basement (Wayne and McCaig, 1998; McCaig et al., 1995; Banks et al., 1991; Bickle et al., 1988) (Fig. 11). Contrarily, rocks from the Mesozoic-Cenozoic cover in the Southern Pyrenean fold and thrust

belt have similar or slightly higher $^{87}Sr/^{86}Sr$ ratios with respect to Phanerozoic seawater (Fig. 11). On the other hand, Cc1a has a narrow range of $\delta^{13}C$, between +0.91 and +2 ‰VPDB, consistent with values of the Devonian marine limestones from the hanging wall (between +1.54 and +2.75 ‰VPDB) and within the range of Devonian marine carbonate values (Veizer et al., 1999). Likewise, the elemental composition of Cc1a follows a trend similar to that of its Devonian host, both having high Mg and Sr and low Mn contents with respect to the other calcite cements (Fig. 12). These geochemical similarities indicate high

fluid-rock interaction and buffering of the carbon and elemental composition of the precipitating fluid by the Devonian carbonates (Marshall, 1992). Calcite cement Cc2 has slightly lower $\delta^{13}C$, lower Mg and Sr and higher Mn contents with respect to both Cc1a and the Devonian host. On the other hand, the temperature and the $\delta^{18}O$ composition of the vein-forming fluids, calculated from clumped isotope thermometry of Cc2, range between 56 °C and 96 °C and between -6.4 and -0.3 ‰SMOW, respectively. These values are interpreted as the involvement of heated meteoric fluids. These fluids, which probably heated

at depth and enriched in radiogenic Sr during their flow through crystalline basement rocks, flowed channelized along the thrust zone (Fig. 13A) due to the enhanced permeability associated with the thrust discontinuity (McCaig et al., 1995; Trincal et al., 2017). As Cc1a and Cc2 precipitated during the same tectonic event but in different structural positions, they likely precipitated from the same fluids, progressively increasing the fluid-rock interaction from the thrust plane (Cc2) towards the





hanging wall (Cc1a). Previous studies already reported channelized syntectonic fluid migration along the thrust zone in other

structures from the Pyrenean basement, such as the Gavarnie thrust and the related Pic-de-Port-Vieux thrust (McCaig et al., 1995).

**5.3 Fluid system during the Neogene extension**

Calcite veins V3 to V5 are attributed to the Neogene extension and the geochemistry of their related calcite cements Cc3 to Cc5 characterize the fluid system during this period.

Cc3 and Cc4 have considerably high $^{87}Sr/^{86}Sr$ ratios (0.714619 and 0.717706, respectively), similar to the ones reported for Cc1a and Cc2 (Fig. 11), indicating interaction with basement rocks. The $\delta^{18}O_{fluid}$ calculated from clumped isotopes, between +4.3 and +12.1 ‰SMOW for Cc3 and between +0.9 and +8.1 ‰SMOW for Cc4, falls within the range of metamorphic and/or formation brines (Taylor, 1987). The $\delta^{18}O$-depleted values of these cements (around -14 ‰VPDB) are due to the high temperatures of the fluids (between 127 °C and 208 °C for Cc3 and between 102 °C and 167 °C for Cc4). Assuming a normal

geothermal gradient of 30 °C, these temperatures would have been reached at a minimum depth of 3 – 5 km. However, these veins have never reached such a burial depth, since during the Neogene extension the studied structure acquired its current configuration (Saura, 2004) and was only buried under the Devonian sequence (hanging wall), which has a maximum thickness of several hundred meters (Mey, 1967). This assumption evidences the hydrothermal character of the circulating fluids. Similarly, the high Mn content of Cc3 and Cc4 (around 7700-8300 and 4000 ppm, respectively), responsible of their bright

luminescence (Fig. 7D, 8C), is consistent with hydrothermal waters (Pfeifer et al., 1988; Pomerol, 1983; Pratt et al., 1991). On the other hand, the $\delta^{13}C$-depleted values of these cements are indicative of the influence of organic derived carbon (Cerling, 1984; Vilasi et al., 2006). The most probable source for these low $\delta^{13}C$ values is the Silurian black shales that do not crop out in the study area but acted as the main detachment level during the Variscan compression, and locally during the Alpine compression, in the Pyrenean Axial Zone (Mey, 1967). These black shales have significant organic carbon contents (TOC

around 2.3%), and around the Gavarnie thrust, these shales have syntectonic carbonate veins yielding $\delta^{13}C$ values between -2 and -8 ‰VPDB (McCaig et al., 1995). Thus, cements Cc3 and Cc4 precipitated from hydrothermal fluids derived and/or equilibrated with crystalline basement rocks and expelled through newly formed and reactivated fault zones during deformation (Fig. 13B).

Finally, the isotopic signature of Cc5, ranging between -8.1 and -5.7 ‰VPDB for $\delta^{18}O$ and between -8.2 and -3.8 ‰VPDB

for $\delta^{13}C$, indicates a meteoric origin. The similar tendency in the elemental composition of this cement and the Stephano-Permian volcanic rocks, both having the highest Mg and Fe and the lowest Mn and Sr contents with respect to the other cements and host rocks, reveals high fluid-rock interaction with the footwall rocks. The significant water-rock interaction is also demonstrated by the Nd isotopic composition of Cc5 (0.512178), yielding to values of the volcanic host (0.512196). This fact, together with the scarcity and small size of Cc5 veins, indicate that this cement precipitated from percolation of meteoric fluids,

which geochemistry was controlled by its volcanic host rock. The $\delta^{18}O_{fluid}$ and the temperature of precipitation calculated from clumped isotopes, ranging between -12.4 and -10.1 ‰SMOW and between -5 and 3 °C, respectively, corroborate the meteoric





origin and may be indicative of high latitude and/or high altitude conditions (Dansgaard, 1964). During the Neogene, the study area was approximately at the same latitude than today (Smith, 1996). Studies focused on infiltration of meteoric fluids and subsequent upflowing along La Tet fault during the Neogene extension, have shown that meteoric waters in the area infiltrate

at high altitudes, around 2000 m, and low temperatures, around 5 °C (Krimissa et al., 1994; Taillefer et al., 2018). On the other hand, the widespread presence of glacial and fluvio-glacial deposits has been reported unconformably overlying the Neogene basin infill and the Variscan rocks from the eastern Axial Pyrenees (Roca, 1996; Turu i Michels and Peña, 2006). These deposits reflect several Quaternary glacial periods in the area, which in turn could have contributed to the low temperature and low $\delta^{18}O_{fluid}$ (Gregory et al., 1989). In any case, precipitation of Cc5 probably took place during the latest stages of extension,

after the fluid regime changed from upward fluid migration to percolation of cold meteoric waters, as also occurred in the Barcelona Plain (Catalan Coastal Ranges) (Cantarero et al., 2014b).

**5.4 Influence of crystalline basement rocks on fluid chemistry during deformation**

The aim of this study was to provide insights into the behavior and evolution of fluids circulating through a long-lived thrust, and to determine the influence of basement rocks on the fluid chemistry during deformation. As pointed above, previous studies

constrained the relationship between deformation and fluid migration in other structures from the Mesozoic and Cenozoic cover (Muñoz-López et al., under review; Cruset et al., 2016, 2018; Travé et al., 1997, 2000), and to a lesser extent, from the Paleozoic basement (Banks et al., 1991; McCaig et al., 1995, 2000b; Wayne and McCaig, 1998). The comparison between these studies and the new data provided in this contribution evidences that fluids migrating through basement or cover units have a different geochemical signature. This signature is recorded in the vein cements, particularly in their $^{87}Sr/^{86}Sr$ ratios.

Accordingly, the high $^{87}Sr/^{86}Sr$ ratios (0.713 to 0.718) of the analyzed calcite cements, originated during successive compressional and extensional tectonic events, indicate that regardless of the origin of the fluids and the tectonic context, basement rocks have a significant influence on the fluid chemistry. This means that cements precipitated from fluids that have circulated through crystalline basement rocks have significantly high $^{87}Sr/^{86}Sr$ ratios (> 0.710) (Fig. 11), reflecting the interaction between the vein-forming fluids with Rb-rich source rocks. Similar high radiogenic $^{87}Sr/^{86}Sr$ ratios have also been

attributed to basement-derived fluids in the Glarus nappe (Swiss Alps) (Burkhard et al., 1992). By contrast, vein cements precipitated from fluids that have circulated through the Mesozoic-Cenozoic sedimentary cover in the Pyrenees have significantly lower $^{87}Sr/^{86}Sr$ ratios (< 0.710). Such lower values may be similar to Phanerozoic seawater values, evidencing interaction between the vein-forming fluids and marine carbonate units, or higher, evidencing interaction with siliciclastic rocks (Cruset et al., 2018; Travé et al., 2007). A previous study, focused on fluid flow along the Gavarnie thrust in the central-

western Axial Pyrenees, used this limit value ($^{87}Sr/^{86}Sr$ = 0.710) to differentiate between the unaltered limestone protolith and the Cretaceous thrust-related carbonate mylonite affected by fluids carrying radiogenic Sr (McCaig et al., 1995).



## 5.5 Fluid flow at regional scale: the NE part of the Iberian Peninsula during the Neogene extension

During the late Oligocene to middle Miocene, the opening of the NW Mediterranean Sea was responsible for the development of a complex ENE-WSW to NE-SW extensional fault system in the NE part of the Iberian Peninsula (Roca, 1996; Roca and
Guimerà, 1992; Vergés et al., 2002). In the eastern Axial Pyrenees, La Tet fault is the main Neogene structure and has associated major E-W extensional faults, such as the Ortedó and La Seu d'Urgell faults, which crosscut the Alpine structures (e.g., the reactivation of the Estamariu thrust) and delimitate the Cerc basin (Fig. 1). Within this basin, E-W mesoscale normal faults (veins V4) also developed during this period and previously formed weakness surfaces were reopened (i.e., the thrust zone foliation associated with the Estamariu thrust, veins V3). During this episode, calcite cements Cc3 and Cc4 precipitated
from hydrothermal fluids (temperatures between 102 and 208 °C) derived and/or interacted at depth with crystalline basement rocks before ascending through newly formed fault zones and reactivated structures. These interpretations are consistent with the presence of several hydrothermal springs (temperatures of 29 °C to 73 °C) currently upwelling aligned along La Tet fault and related Neogene deformation in the Pyrenean Axial Zone (Krimissa et al., 1994; Taillefer et al., 2017, 2018). Several studies indicate the origin of these hot water springs as meteoric fluids, infiltrated at high-elevated reliefs, above 2000 m,
warmed at great depths by normal geothermal gradients, and migrated upwards along permeability anisotropies related to fault zones (Taillefer et al., 2017, 2018). The geochemical analysis of these springs, and in particular their high radiogenic $^{87}$Sr/$^{86}$Sr ratios, ranging between 0.715 and 0.730, according to French Geological Survey reports (Caballero et al., 2012), are within the range of values obtained in this study and also accounts for interaction between circulating fluids and crystalline basement rocks. Studies based on numerical models suggest that La Tet fault and the involved crystalline rocks are still permeable down
to 3 km depth (Taillefer et al., 2017, 2018), although the fault has been dormant since the Mio-Pliocene (Goula et al., 1999). These authors also suggest that the footwall topography is the major factor controlling the infiltration of meteoric fluids and the recharge of the hydrothermal system. The topography, which induces high hydraulic gradients and produces fluid advection, controls the circulation depth and therefore, the maximum temperature reached by the migrating fluids (Taillefer et al., 2017).
A similar geological context and fluid regime evolution to that explained above is found in the Barcelona Plain and the Vallès Basin, located in the northeast part of the Catalan Coastal Ranges (CCR) (Fig. 1A). Consequently, the comparison between both geological contexts allows us to give insights into the fluid circulation in extensional basins at regional scale (in the NE part of Iberia). In these locations of the CCR, the main fault system associated with the Neogene extension acted as conduits for hydrothermal fluid circulation at temperatures between 130 and 150 °C during synkinematic periods (Cantarero et al.,
2014a, 2014b; Cardellach et al., 2002), and is also responsible for the present-day circulation of hot water springs up to 70 °C (Carmona et al., 2000; Fernàndez and Banda, 1990). In both cases, fluids would have been topographically driven from elevated areas to great depths (Cantarero et al., 2014b), where they circulated through crystalline rocks, acquiring high $^{87}$Sr/$^{86}$Sr ratios (> 0.712) and high temperatures (Cardellach et al., 2002) before ascending through fault discontinuities. However, in the Penedès basin, which corresponds to the southwestern termination of the Neogene structure in the CCR, crystalline




basement rocks do not crop out and the extensional faults only involve Neogene deposits filling the basin and a Mesozoic sedimentary substrate. In this location, the main fault system acted as conduits for several episodes of meteoric fluids percolation during the Neogene extension and evidence of hydrothermal fluid circulation has not been reported in the area (Travé and Calvet, 2001; Travé et al., 1998; Baqués et al., 2012, 2010). This fact agrees with previous studies that highlight that hydrothermal activity, and in particular the occurrence of hot water springs in the Pyrenees and in the CCR, is preferably concentrated in basement rocks, which constitute the elevated footwall of the main extensional fault systems (Taillefer et al., 2017; Carmona et al., 2000).

All these observations indicate that the fluids responsible for precipitation of synkinematic cements during Neogene times in the eastern Axial Pyrenees and in the northeast part of the CCR, and fluids currently flowing through Neogene extensional faults in both places are hydrothermal and have crystalline basement rocks as a common reservoir. This evidences an open fluid system in the NE part of the Iberian Peninsula associated with the Neogene extensional deformation. Accordingly, this extensional fault system has acted as a conduit for long-term circulation of hot fluids from Neogene times to present. This long-term fault-controlled fluid flow could have been continuous through time or could be related to intermittent pulses. Fault control on upflowing of hot fluids along fault systems is a common process in different geological settings and has been reported in the Great Basin, USA (Faulds et al., 2010), in the Western Turkey (Faulds et al., 2010), in the Southern Canadian Cordillera (Grasby and Hutcheon, 2001), in the Basin and Range Province (Nelson et al., 2006) and in the southern Tuscany, Italy (Liotta et al., 2010).

## 6 Conclusions

The Estamariu thrust, in the Pyrenean Axial Zone, resulted from a multistage Variscan to Neogene tectonic evolution. Our data, combining structural and petrological observations with geochemical analyses of synkinematic calcite veins and host rocks, provide a structural and diagenetic framework that constrains the fault-fluid system evolution and assesses the relationships between deformation and fluid migration in the Pyrenean basement. In the study area, the Variscan Estamariu thrust places a Devonian pre-Variscan unit against a Stephano-Permian late to post-Variscan sequence and therefore, the structures present within the thrust zone, affecting both sequences, are attributed to the Alpine and subsequent Neogene reactivation of the thrust. During the Alpine compression, the reactivation of the thrust resulted in the transposition of the Variscan regional foliation within the thrust zone and in the formation of a subsidiary thrust zone affecting the andesites in the footwall. During this period, meteoric fluids interacted with the crystalline basement and migrated upwards along the thrust and related structures at temperatures between 56 and 98 ºC. These fluids progressively increased the fluid-rock interaction from the thrust plane towards the hanging wall. During the Neogene extension, the Estamariu thrust was reactivated and normal faults and shear fractures were formed. These structures allowed basement-derived fluids to flow upwards through reactivated and newly formed fault zones at temperatures up to 208 ºC. Finally, during the latest to post stages of extension and uplift of



the structure, the fluid regime changed to percolation of low temperature meteoric fluids that were buffered by the volcanic host rocks.

The comparison between our results and previously published data allows us to provide insights into the fluid characteristics and fluid regime at regional scale. On the one hand, the influence of basement rocks on the fluid chemistry during deformation

in the Pyrenees has been assessed. In this sense, regardless the fluid origin and the tectonic context, the fluids that have interacted with crystalline basement rocks have a significantly higher $^{87}Sr/^{86}Sr$ ratio ($> 0.710$) with respect to those that have circulated through the sedimentary cover ($< 0.710$). On the other hand, a similar fluid regime associated with the Neogene extension in the NE part of the Iberian Peninsula (including the eastern Pyrenees and the northeastern part of the Catalan Coastal Ranges) has been observed. In both settings, the extensional deformation structures have acted as conduits for long-

term fluid migration from Neogene times to present. Migrating fluids during this period are hydrothermal and have interacted with crystalline rocks before ascending through fault zones and related structures.

**Acknowledgments**

Carbon and oxygen isotopic analyses were carried out at "Centre Científics i Tecnològics" of the Universitat de Barcelona. Strontium and Neodymium analyses were performed at the "CAI de Geocronología y Geoquímica Isotópica" of the

Universidad Complutense de Madrid. The Elemental composition was analysed at the Geochemistry Facility of labGEOTOP of the Institute of Earth Sciences Jaume Almera (ICTJA-CSIC). Clumped isotope thermometry was carried out at the Imperial College London. This research was carried out within the framework of DGICYT Spanish Project PGC2018-093903-B-C22 (Ministerio de Ciencia, Innovación y Universidades / Agencia Estatal de Investigación / Fondo Europeo de Desarrollo Regional, Unión Europea) and the Grup Consolidat de Recerca "Geologia Sedimentària" (2017-SGR- 824). The PhD research

of DM-L is supported by the FPI2016 (BES-2016-077214) Spanish program from MINECO.

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





**Table 1. δ¹⁸O, δ¹³C, ⁸⁷Sr/⁸⁶Sr and ¹⁴³Nd/¹⁴⁴Nd ratios of the calcite cements and related host rocks. The calculated precipitation**
**temperature and the δ¹⁸O$_{fluid}$ of the parent fluids are also indicated. NR indicates analyzed samples in which no result was obtained.**

| Sample | Vein | Cement | $\delta^{18}O$ ‰VPDB | $\delta^{13}C$ ‰VPDB | $^{87}Sr/^{86}Sr$ | $^{143}Nd/^{144}Nd$ | $\Delta_{47}$ | T (°C) | $\delta^{18}O_{fluid}$ ‰SMOW |
|---|---|---|---|---|---|---|---|---|---|
| C9 | V1a | Cc1a | -11.2 | +0.91 | | | | | |
| C8B | V1a | Cc1a | -10.7 | +2 | | | | | |
| C8A.I | V1a | Cc1a | -10.4 | +2 | | | | | |
| C8A.II | V1a | Cc1a | -10.96 | +1.3 | 0.713018 | NR | | | |
| C8A.III | V1a | Cc1a | -10.9 | +1.2 | | | | | |
| C7.I | V1a | Cc1a | -10.9 | +2.1 | | | | | |
| C7.II | V1a | Cc1a | -10.8 | +0.8 | | | | | |
| C7.III | V1a | Cc1a | -10.4 | +1.96 | | | | | |
| C4B | V1a | Cc1a | -10.3 | +1.9 | 0.714092 | NR | | | |
| C3A.I | V1a | Cc1a | -11.2 | +1.9 | | | | | |
| C3A.II | V1a | Cc1a | -11.3 | +1.7 | | | | | |
| C3A.III | V1a | Cc1a | -10.5 | +1.98 | | | | | |
| C15.I | V2 | Cc2 | -14.9 | -1.2 | 0.718294 | NR | 0.567 | 56 to 98 | -0.3 to -6.4 |
| C15.II | V2 | Cc2 | -13.3 | +0.5 | | | | | |
| C15.III | V2 | Cc2 | -12.91 | +1.54 | | | | | |
| C13 | V3 | Cc3 | -13.8 | -7.1 | 0.714619 | NR | 0.445 | 127 to 208 | +4.3 to +12.1 |
| C12.II | V3 | Cc3 | -14.3 | -7.3 | | | | | |
| C10 | V3 | Cc3 | -14.2 | -9.3 | | | | | |
| C11A | V3 | Cc3 | -14.2 | -8.7 | | | | | |
| C13.II | V3 | Cc3 | -13.6 | -7.2 | | | | | |
| C14.I | V3 | Cc3 | -13.4 | -6.9 | | | | | |
| C14.II | V3 | Cc3 | -13.7 | -7.4 | | | | | |
| C16A | V3 | Cc3 | -13.8 | -7.2 | | | | | |
| C16B | V3 | Cc3 | -14 | -7 | | | | | |
| C16C | V3 | Cc3 | -14.1 | -6.9 | | | | | |
| C18.I | V4 | Cc4 | -13.4 | -7.2 | 0.717706 | | 0.48 | 102 to 167 | +0.9 to +8.1 |
| C18.II | V4 | Cc4 | -13.8 | -7.4 | | | | | |
| C12.I | V5 | Cc5 | -8.1 | -7.8 | 0.716923 | 0.512178 | | | |
| C6.I | V5 | Cc5 | -6.7 | -8.2 | | | | | |
| C6.II | V5 | Cc5 | -7.4 | -7.4 | | | | | |
| C11B | V5 | Cc5 | ;.7 | -3.8 | | | | | |
| C3A.HR | Devonian carbonates | | ).5 | +2.4 | 0.710663 | NR | 0.77 | -5 to +3 | -12.4 to -10.1 |
| C17.HR | | | ).51 | +1.54 | | | | | |
| C4.HR | | | .36 | +2.7 | | | | | |
| C11.HR | Andesites | | - | - | 0.743983 | 0.512196 | | | |






**Table 2: Elemental composition (Ca, Mg, Fe, Mn, Sr) of the calcite cements Cc1a to Cc5 and host rocks from the hanging wall (HW) and footwall (FW). The scale in greens indicates higher concentrations when is darker.**

| Sample | Ca ppm | Mg ppm | Fe ppm | Mn ppm | Sr ppm |
|--------|--------|--------|--------|--------|--------|
| Cc1a | 391618 | 1335.9 | 5603.5 | 1243.7 | 543.7 |
| Cc1a | 349063 | 1548.6 | 4121.3 | 781.6 | 460.2 |
| Cc1a | 351134 | 1231.8 | 5205.4 | 810.3 | 545.5 |
| Cc1a | 337588 | 1126.6 | 3914.9 | 680.7 | 704.0 |
| Cc2 | 328169 | 501.2 | 1061.4 | 3629.3 | 248.5 |
| Cc3 | 364995 | 331.9 | 1647.8 | 8277.9 | 122.3 |
| Cc3 | 333123 | 909.5 | 5545.9 | 7695.5 | 424.9 |
| Cc4 | 333563 | 624.4 | 3814.9 | 4034.6 | 72.2 |
| Cc5 | 233784 | 2260.0 | 8656.6 | 161.4 | 72.1 |
| Cc5 | 281741 | 1626.2 | 4078.0 | 138.8 | 25.3 |
| HW | 320038 | 2752.6 | 6289.0 | 621.1 | 449.4 |
| FW | 4234 | 12830.5 | 43107.1 | 466.6 | 18.6 |






**Figure 1: (A)** Simplified geological map of the Pyrenees modified from (Muñoz, 2017) and its location in the Iberian Peninsula (location of the Catalan Coastal Range, CCR, is also shown). **(B)** Detail of the study area located within the Pyrenean Axial zone.
**(C)** Geological map of the Cerc basin (using data from Saura (2004) and our own data) with the Estamariu thrust located in its eastern termination and the Neogene extensional faults in the northern and southern limits. The white square indicates the location of the main outcrop (Fig. 2A).





**Figure 2: (A)** Geological map and **(B)** cross-section of the Estamariu thrust, which juxtaposes a Devonian unit against a Stephano-
Permian sequence (H=V, no vertical exaggeration). Lower-hemisphere equal-area stereoplots of the Devonian bedding (S0), regional
foliation (Sr), thrust zone foliation affecting the hanging wall (SD) and footwall (SSP), magmatic layering (Sm) and the different
faults and veins observed in the study area are also included. Location in Fig. 1B.





**Figure 3: Sketch showing the spatial distribution of mesoscale structures within the main outcrop and lower-hemisphere equal-area stereoplots of the different mesostructures.**



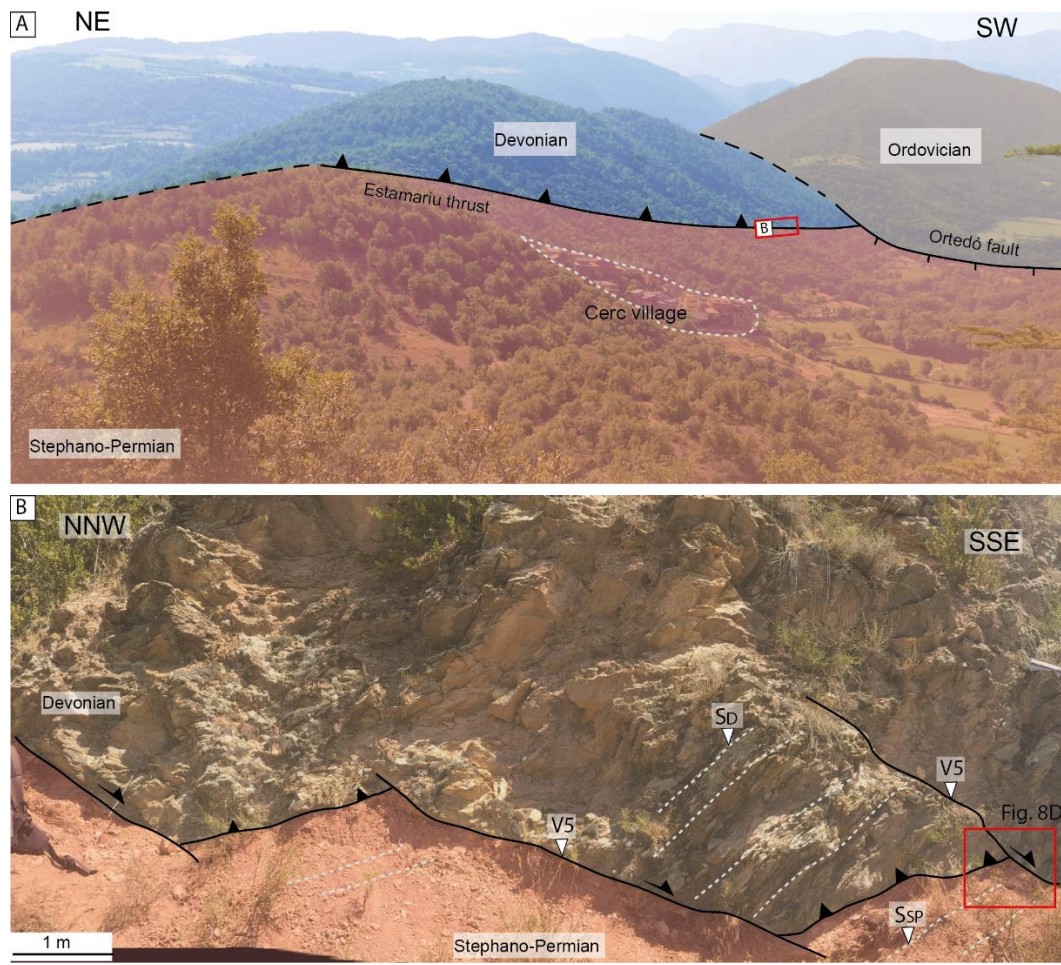

**Figure 4: Main outcrop of the Estamariu thrust. A) Panoramic view from the Sant Antoni hill showing the extensional Ortedó
fault postdating the Estamariu thrust. B) Main outcrop showing the Estamariu thrust and the related foliation developed in the
Devonian hanging wall (SD) and in the Stephano-Permian footwall (SSP). The thrust is displaced by later shear fractures V5.**



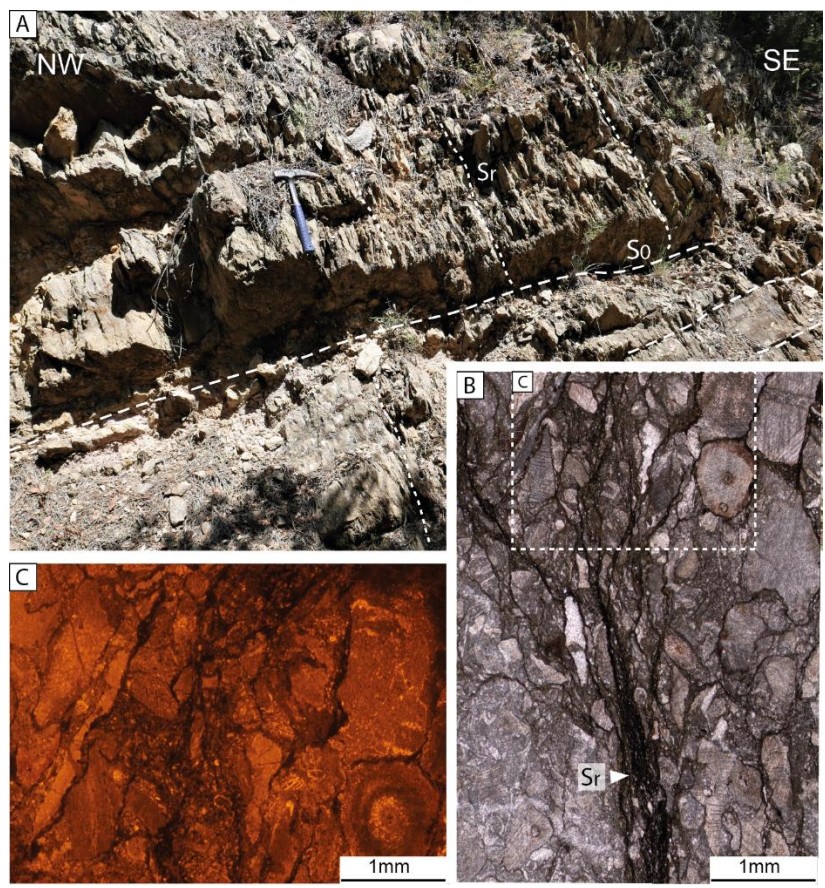

**Figure 5: Devonian protolith. A) Field image showing the relationship between bedding ($S_0$) and regional foliation (Sr). B) Plane polarized light and C) Cathodoluminescence microphotographs of the encrinites alternating with pelitic rich bands, where the Sr is concentrated.**





**Figure 6: Mesostructures and microstructures found within the thrust zone affecting the hanging wall. A) Outcrop image of the thrust zone foliation ($S_D$) and related C planes indicating reverse kinematics (Ci). Microphotographs of B) thrust zone foliation, C) stylolites e1 and veins V0 affecting the Devonian encrinites, D) Crossed polarized light and E) cathodoluminescence microphotographs of veins V1a concentrated between foliation surfaces. F) Thrust zone foliation ($S_D$) near the fault plane and ambiguous and perpendicular relationships between V1b and e2. G) Crossed polarized light and H) cathodoluminescence microphotographs of calcite cement Cc2 located on the main thrust plane (V2).**




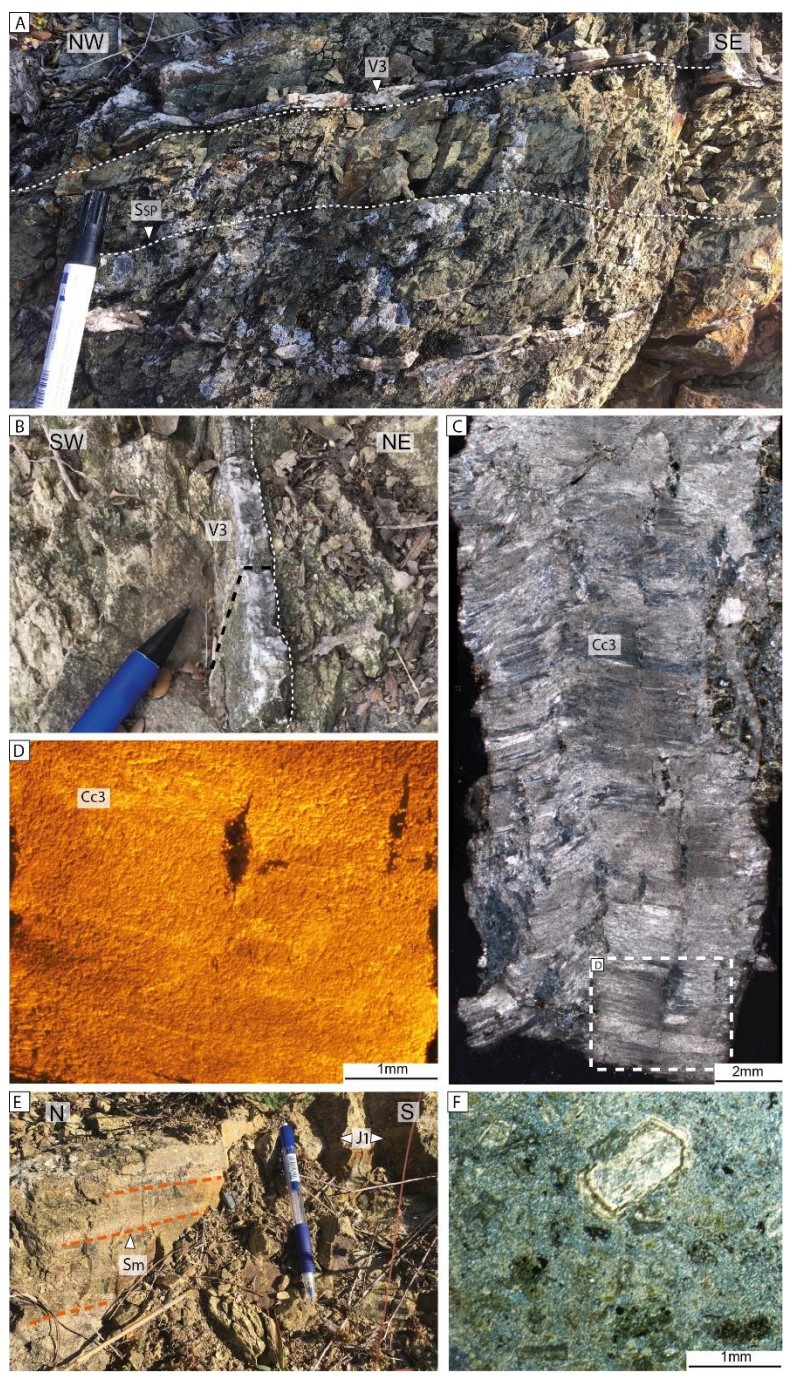

**Figure 7: Mesostructures and microstructures present in the Stephano-Permian volcanic footwall. A) Field image of the subsidiary thrust zone in the footwall showing the thrust zone foliation (SsP) and the veins V3. B) Detail of veins V3 also in the subsidiary thrust zone. The black dashed line indicates the original position of the thin section observed in C. C) Crossed polarized light and D) cathodoluminescence microphotographs of veins V3, characterized by calcite fibers growing perpendicular to the vein walls (Cc3). E) Field image of the footwall andesites showing the magmatic layering (Sm) and joints J1. F) Plane polarized light microphotograph of the volcanic andesites exhibiting a porphyritic texture with a large plagioclase crystal.**






**Figure 8: A) Field image of a subvertical and E-W fault plane mineralized with calcite (veins V4) and showing two striae set generations (white arrows) indicating dip-slip and strike-slip kinematics. B) Crossed polarized light and C) cathodoluminescence microphotographs of the vein-related calcite cement (Cc4). D) Shear fracture postdating the thrust zone foliation, locally mineralized with calcite veins V5. E) Shear bands (Cn) with normal kinematics located in the main thrust zone, indicating a later reactivation of the Estamariu thrust.**





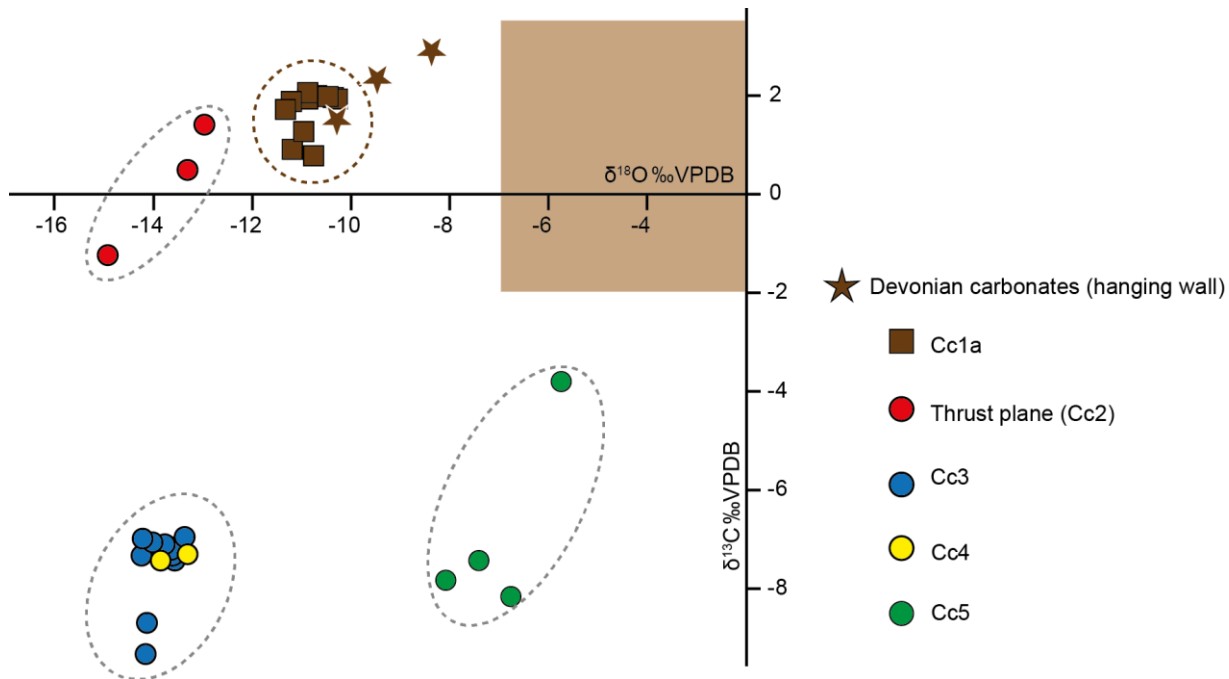

**Figure 9: δ¹⁸O and δ¹³C values of calcite cements Cc1a to Cc5 and the Devonian carbonates from the hanging wall. The brown box refers to typical values of Devonian marine carbonates (Veizer et al., 1999).**







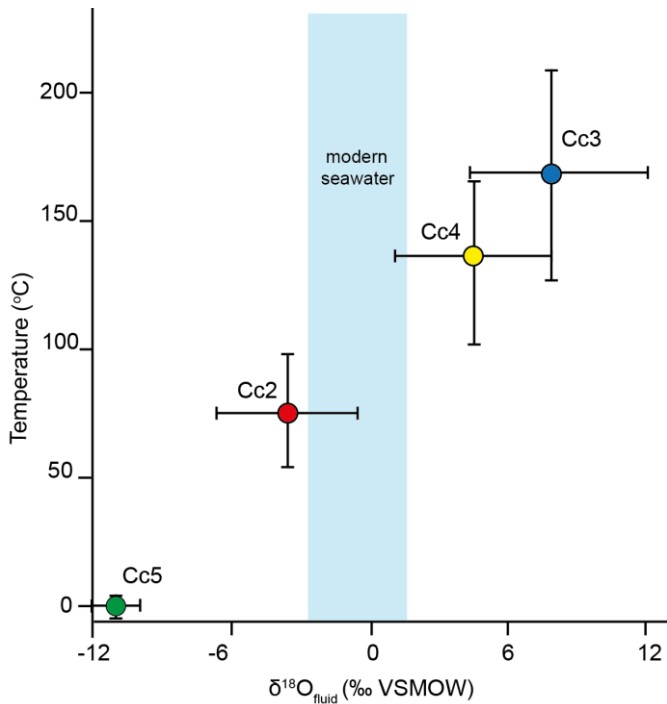


**Figure 10: Temperatures (°C) vs δ$^{18}$O$_{fluid}$ calculated for cements Cc2 to Cc5. The typical δ$^{18}$O values for modern seawater (blue band) are from Veizer et al., (1999).**





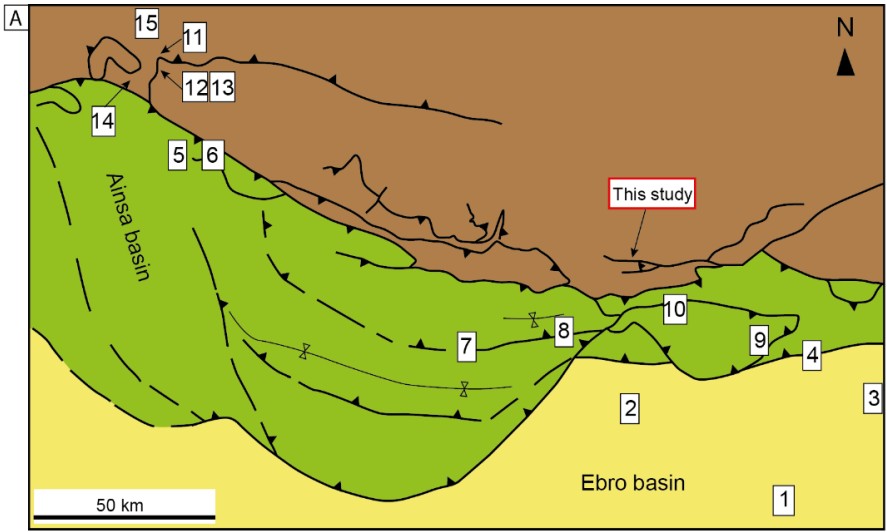

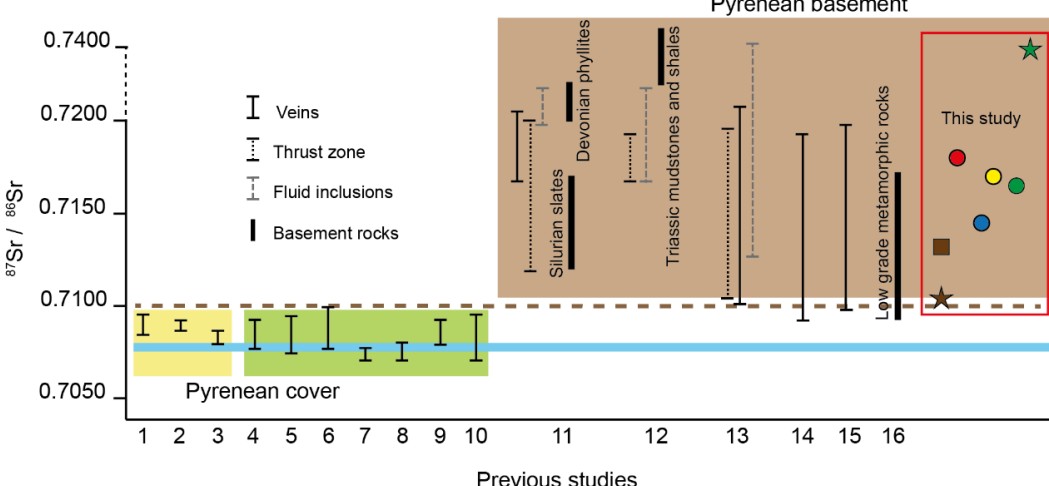


**Figure 11: Simplified geological map of the south-central Pyrenees showing the location of structures where $^{87}$Sr/$^{86}$Sr analysis have been carried out. Below, $^{87}$Sr/$^{86}$Sr ratios from this study compared to results from other structures involving either cover units (1-10) or basement rocks (11-16). The blue thick line refers to the $^{87}$Sr/$^{86}$Sr range of Phanerozoic seawater and the dashed brown line represents the $^{87}$Sr/$^{86}$Sr limit value between basement and cover structures. 1. El Guix anticline (Travé et al., 2000), 2. Puig Reig anticline (Cruset et al., 2016), 3. L'Escala thrust (Cruset et al., 2018), 4.Vallfogona thrust (Cruset et al., 2018), 5. Ainsa basin (Travé et al., 1997), 6. Ainsa-Bielsa area (McCaig et al., 1995), 7. Minor Bóixols thrust (Muñoz-López et al., under review), 8. Bóixols anticline (Nardini et al., 2019), 9. Lower Pedraforca thrust (Cruset, 2019), 10. Upper Pedraforca thrust (Cruset, 2019), 11. Gavarnie thrust (McCaig et al., 1995), 12. Pic de Port Vieux thrust (Banks et al., 1991), 13. Pic de Port Vieux thrust (McCaig et al., 2000b), 14. Plan de Larri thrust (McCaig et al., 1995), 15. La Glere shear zone (Wayne and McCaig, 1998). 16. Trois Seigneurs Massif (not in the map) (Bickle et al., 1988).**





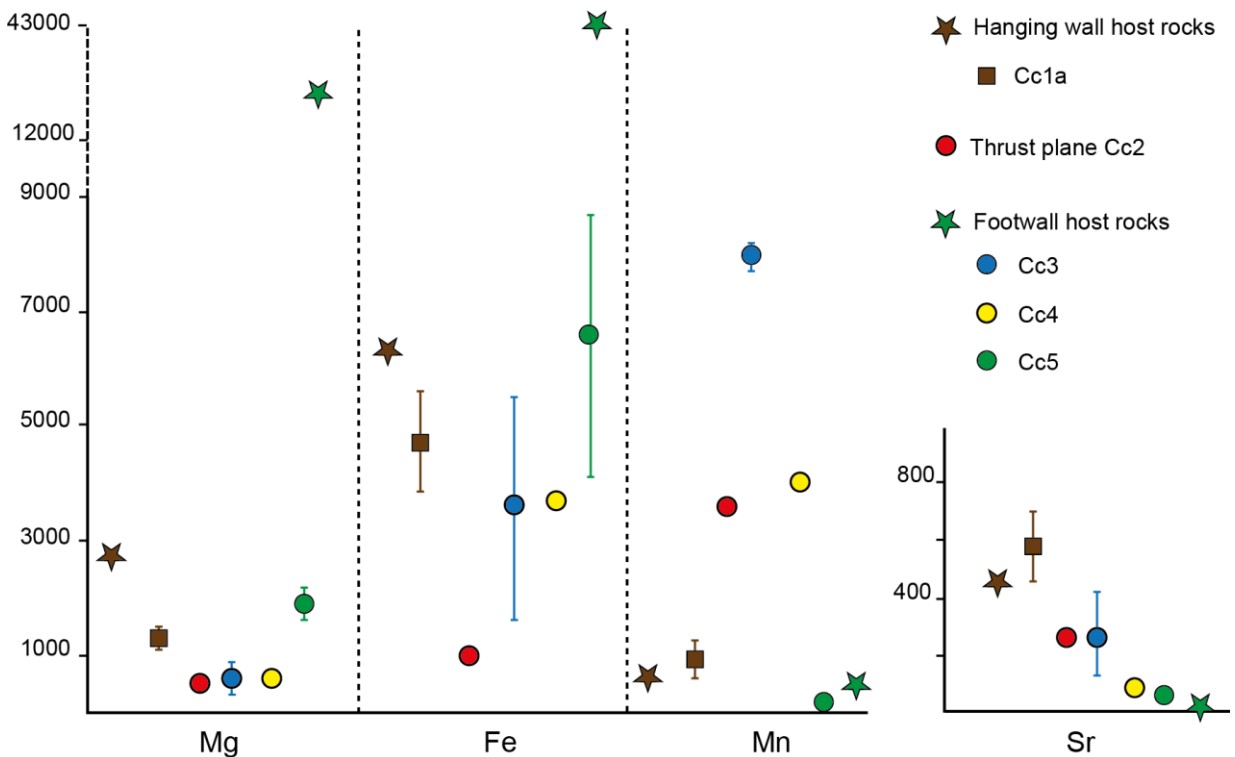

**Figure 12: Elemental composition (including Mg, Fe, Mn and Sr) in ppm of the different calcite cements and host rocks. Bars**
**indicate maximum, minimum and average composition.**





**Figure 13: Tectonic evolution of the study area (not to scale) and relationship with the evolution of the fluid system. A) During the Alpine reactivation of the Estamariu thrust, a meteoric fluid (yellow arrows) interacted at depth with basement rocks and then migrated channelized along the fault plane towards the hanging wall, precipitating cements Cc1a and Cc2. B) During the Neogene extension, basement-derived hydrothermal fluids (blue arrows) flowed upwards through newly formed and reactivated fault zones. This fluid precipitated calcite cements Cc3 and Cc4. Finally, during ongoing deformation, meteoric fluids (green arrows) percolated in the system and precipitated Cc5.**