# Peer review of "Influence of basement rocks on fluid evolution during multiphase deformation: the example of the Estamariu thrust in the Pyrenean Axial Zone"

_Solid Earth, 2020_

## Referee Comment (RC1) · Owen Callahan (Referee) · 7 Jul 2020

Review for Muñoz-López et al. "Influence of basement rocks on fluid evolution during multiphase deformation: the example of the Estamariu thrust in the Pyrenean Axial Zone" submitted by Owen Callahan

Overview: Muñoz-López and co-authors document evolving fluid sources and conditions during multiple stages of deformation through a combination of field, petrographic, and geochemical analysis of an exhumed thrust in the Pyrenees. The manuscript is

generally well written, with only minor grammatical or stylistic edits, and the quality of the work is robust. I do think that the manuscript could be improved with relatively minor edits, specifically with regards to motivation, descriptions of geologic units, and inclusion of a better synthesis figure.

General Comments: The motivation for the study should be stated clearly and much earlier in the document. The clearest iteration that I found was on page 13, the first sentence of section 5.4. It would be helpful for this type of statement to appear in the abstract, and in the introduction.

For all the discussion of basement, the age, lithology, and metamorphic grade were only briefly discussed, and relatively late in the text. It would be helpful to expand and highlight the section on the basement geology, and specifically previously published geochemistry, instead of the broader regional tectonism. It is particularly confusing because you repeatedly refer to "crystalline basement" but we learn that the basement is composed of Paleozoic slates, phyllites, sandstone, mudstone, limestone, conglomerate, and shale in Figure 11 and at the very end of the background geology, and if these are not the basement rocks then I'm completely lost. This is perhaps relevant because the Sr data is really more about interaction with older rocks with a different radiogenic signature, not rocks that are "crystalline" or not, correct?

I would like to see a figure of chemistry/temperature/source and deformation/tectonic events through time, rather than the 3D block diagram. The source and direction of infiltration and upwelling are rather speculative, but putting your observations into a more linear, temporal framework would be really helpful for me (see for instance my quick quick sketch). Similarly, annotating the figures showing geochemistry with some additional information about the inferred timing of cements, or associated structures, would be very helpful.

What you meant by "timing" was a little misleading. At first, I assumed you were talking about relative timing from cross cutting relationships, but then you mention radiogenic

ages, but then you acknowledge that you did not get ages. . . I think it would be better to be more specific, and upfront, about these being relative ages that are broadly linked to specific styles of tectonism.

This was originally a specific comment, but I think it applies to the whole manuscript: Many hydrothermal systems are ultimately derived from meteoric fluids, and I agree that there is a clear and reasonable distinction between your cooler, younger fluids and hotter fluids, but a short sentence relating the importance of fluid-rock reaction and chemical evolution at higher temperatures and longer times would be helpful, because I think you are really making the point that you can track fluids because of that residence time at depth, not their ultimate source.

Specific Comments:

Line 13: "timing" is a bit vague here. Because absolute age in faults and fractures is such a hot topic, I think it is important to specifically describe relative ages throughout.

Lines 17-21: Crystalline basement is vague, especially in light of the abundance of low grade and unmetamorphosed rocks you show in later figures. Also, the distinction between deeply circulated meteoric and hydrothermal fluids. . . Is this just a matter of temperature or chemical evolution?

Line 33: I'm not sure how understanding past fluid flow helps use understand the current configuration of a mountain belt. Perhaps it informs the factors leading to the current configuration? Additionally, I think there is a missing comma after "through time"

Line 51: You say only a few, but cite 7 papers working on similar topics. Also, "On the other hand" does not seem necessary here.

Line 100: These are the crystalline basement rocks? I think the geologic background should be more clear about rock types much earlier.

Line 105: I'd like to see a short description of why samples were selected before

launching into all the analysis, i.e. the field methods component, which is well documented in your figures. You include a lot of structural context in "Results" but it is difficult to evaluate whether 35, 12, or 8 samples is enough without some description of the structures that are present.

Line 121: Extra comma in "then, they"

Line 138: Extra comma in "one, keeping"

Line 162: First mention of U-Pb geochronology. So is this about absolute age?

Line 191: I understand the shorthand for regional foliation being Sr, but it is unfortunate that this paper also discusses strontium. Perhaps use Sr?

Line 241: Perhaps "steeply" dipping?

Line 246: "frequently affect"? Commonly may be a better choice; frequent implies some element of time.

Lines 225-253: It may be imbedded in the figures, or I may have been tired at this point, but I felt like the cross cutting relations or structural context for sequencing was a bit week with veins and cements 3-5.

Lines 262-266: This could go in a table.

Line 301: Oh, there is NOT absolute geochronology. This passage should appear much earlier.

Line 304: An example of the basement lithologies and ages being clearly and simply defined.

Line 330. Capitalize "Calcite..."

Lines 355-361: Emphasis on channelized is a little confusing, because you also say that cements in different structural positions precipitated from the same fluids, which then begs the questions How wide are the fluid flow channels? Do you have samples

from inside and outside of these channels? I think this invites a lot of extra scrutiny, and as it is written it is too vague.

Lines ~385-390: I think you do a good job defending your interpretation of fluid sources here.

Line 393: replace "than" with "as"?

Line 402: Probably the clearest statement of purpose in the manuscript.

Line 464: "Common reservoir" implies a system in hydrologic or pressure communication. I don't think this is necessarily supported. Rather, you could claim that fluids are sourced or resided in similar basement rocks.

Lines 469-470: Great Basin and Basin and Range Province are not necessarily synonymous, but they are in this case and therefore redundant.

Line 495: Although you do include a caveat, I think the phrase "long-term" implies persistence, which is not necessarily true. I think places in the text that you describe long-term fluid flow should be revisited.

Line 790 (Table 1): alignment issue after sample C6.II in some columns impacts readability. Also check on reason for differences in significant digits for reported values of the same quantity (for instance, see d18O and d13C columns)

Line 800 (Table 2): check grammar "when is darker". A scale bar for color might be useful so we know what is considered high or low or how divisions were made (are they quartiles, or relative to some standard?)

Line 805 (Figure 1): Including generic rock types instead of or in addition to ages would be helpful, see generic comment about describing the lithology of the basement rocks.

Line 855 (Figure 9): The relative timing of cements can be inferred from their order, but perhaps arrows on the graph showing fluid evolution over time, or some other way to relate these to types of structures or tectonic events would be useful. See general

comment about a timeline figure.

Line 865 (Figure 10): Again, this could be combined with a figure showing fluid evolution in the context of other events or structures, rather than leaving it to the reader to relate samples and setting. Add more text on the graphs to help guide me.

Line 874 (Figure 11). Shales and mudstones don't seem particularly "crystalline", but you refer to crystalline basement a few times in the manuscript. I can understand why dilatant crystalline rocks may be geothermal reservoirs, but why these low grade rocks? Is it just that these rocks exist at deeper, hotter depths? Brown line seems a bit high (it cuts off a few basement values), how was it chosen?

Line 885 (Figure 12). Again, I'd rather follow the changes over time, so instead of looking at changes in Mg, then Fe, then Mn. . . plot these values in the context of other events and features and then we could see what was happening with Mg when Sr goes up, for instance.

Line 890 (Figure 13). This is a fine figure, but I don't know that it adds a great deal to the story, other than showing what you have already described fairly well in the text. If you do keep it, I think showing warmer colors for hotter fluids may be more intuitive.
* * *
[Figure]

**Fig. 1.**

---

## Referee Comment (RC2) · Brice LACROIX (Referee) · 31 Jul 2020

This paper by Munoz-Lopez et al. reports structural and geochemical evidence of several multi-phase fluid-flow along the Estamariu thrust located in the Pyrenean Axial Zone combining detailed structural and microstructural observation to O, C, Sr and $\Delta 47$ isotopes. The techniques employed here are adequate and I'm glad to see application of $\Delta 47$ thermometry in the Axial zone. The dataset is sound, and the conclusions are reasonable. The authors document a complex fluid-flow history along the thrust during Alpine compression and Neogene extension involving different sources of fluid

at different temperatures. This paper should be published as it is an important regional contribution. As stated by the authors, there is a long list of works studying the fluid-flow along thrust faults affecting the sedimentary cover (e.g. Southern Pyrenean zone), but just a few focuses on the basement from the Axial Zone.

Although this work should be published, there are several points that need to be addressed/commented during revision.

1. The structural analysis and description are very detailed but somehow confusing. Here are some suggestions that could clarify the description: - Adding sub-titles such as 4.1.1. Adding sub-title such as Hanging wall, Main Thrust, and Footwall would help a lot.

- I would also recommend adding a general schema synthetizing all the relationship between the observed structures and microstructures. The current Figure 3 does that, but it is still confusing;

- Also, I found the adopted typologies for structures "Sr, Sm, Ssp, SD, etc. . ." confusing and did not catch up what the subscript letter ('r' and 'm') refer to. What about calling these foliations the same way (unless they are associated to different tectonic phases) and just describe them (morphologies, intensity, orientation) in the sub-section. This would simplify understanding of the numerous stereoplots presented in Figure 3. I also noticed that some of these foliation typologies are not called in the text. For example in line 199: "The foliation within the thrust zone affecting the Devonian hanging wall strikes NW-SE and dips 40 – 50o NE, similar to the regional foliation in the protolith, but it is more closely spaced, generally between 0.2 and 1 cm (Fig. 6A, B)." Specify in the text if this foliation corresponds to "Sr". The same comment is applicable for all the section 4.1.

- You mentioned pressure solution surfaces e1 and e2 but it was not clear on which basis they were differentiated. Is there any cross-cutting feature? The orientation of these features (if any) are not presented on stereoplots.

2. The authors report warm temperature fluid-flow event (up to > 200ËŽC), presumably hydrothermal fluid for the CC3 and CC4 calcite phases. As you know, $\Delta47$ composition of carbonate may be altered by $\Delta47$-reordering when carbonate experienced temperature in excess of 200ËŽC (maybe lower temperature). Although, I'm convinced that these hydrothermal events did not altered previous carbonate phases (cc1, cc2), I would like to see the authors discussing potential (or not) $\Delta47$-reordering and how it could be ruled out. Here are some ways to discuss that: - Is there some metamorphic/Fluid Inclusion/chronology work in the same area reporting temperature-time relationship of this hydrothermal event? If this hydrothermal event if short, the solid-state $\Delta47$ probably did not occur; - Is the thermal history of the area constrained by other studies? If it is the case, the authors could use re-ordering models (e.g. Stolper and Eiler 2015; Lloyd et al., 2017) to see if the $\Delta47$ composition of the different calcite phases may experience re-ordering. - Alternatively, the authors may acknowledge that further re-ordering is possible but unlikely due to the short time.

3. I have noticed few poorly constructed sentences. I would suggest the English to be checked before re-submission. I won't make any comment on that as I'm myself always struggling with English.

Minor comments: l. 28: Deformation associated with crustal shortening is mainly accommodated by thrust faulting and related fault zone structures: Add references. l. 30: "The reactivation of faults may produce changes in the hydraulic. . ..": Add References. l. 174: Which fractionation curve is used to calculate the oxygen isotope composition of water? l.180 – 181: "The main slip plane is undulose, producing changes in the strike direction and dip, and generates a 2 – 3 m thick thrust zone, which is thicker in the hanging wall, up to 2.5 m thick": Do you mean the thrust fault consists to a deformation zone affecting both hanging wall and footwall, with deformation zone thicker in the HW? l.187: "In the studied outcrops, the Devonian Rueda Formation from the hanging wall is characterized by a well-bedded alternation of dark to light grey limestones with dark grey shales": Does this refer to S0, Sr, Sd? Please specify.

l.190-192: "Deformation in the Devonian protolith (i.e., outside the thrust zone) corresponds to a decametric anticline (Fig. 2B), which has associated an axial plane pervasive regional foliation (Sr) concentrated in the pelitic intervals (Fig. 5B)": looking at the stereoplot from Fig. 3, the bedding (S0) seems to define a fold oriented E-W (although only based on 3 measurements). In contrast the Sr does not seem parallel to the axial plane and is more or less oriented N-S (even slightly folded). How can you explain this?

l. 203-204: "When present, these stylolites are very systematic with densities between 5 and 8 stylolites/cm." Should the intensity be given in number/cm2?

l. 201 – 202: "At mesoscale, SD has related shear surfaces (Ci) defining centimetric S-C-type structures, again indicating reverse kinematics (Fig. 6A).": Do you have a closer view of the C-S relationship?

l.220-222: "The vein cement (Cc2) is milky white in hand sample and consists of up to 3 mm blocky to elongated blocky crystals (Fig. 6G) with a dull to bright orange luminescence (Fig. 6H).": To me, e2 and V1b are clearly cogenetic as their crosscutting relationship are ambiguous (as stated by the authors). It is also interesting to see that V1b is extensional (Mode I) but also show mode II with conjugate opening (Fig. 6F). In any case all these structures formed under the same field stress and can be assumed contemporaneous.

l.226: "They are parallel or locally branch off cutting the foliation planes in the subsidiary thrust zone": What are the textures of these veins? They seem to show interdigitated texture in agreement with extension opening. These are important as they give indication of the opening regime and stress field.

l.246: You state here v4 for fault. However, you previously used V labels for veins. It is confusing even if we expect slickenside onto these faults.

l.252-253: "Shear fractures (V5) are locally mineralized with a greyish microsparite

calcite cement (Cc5).": Any Figure to document?

l. 400: Huyghe et al. (2018): "Impact of topography, climate and moisture sources on isotopic composition ($\delta$18O & $\delta$D) of rivers in the Pyrenees: Implications for topographic reconstructions in small orogens" reported new isotope lapse rates for the Pyrenees. This study should be cited here as it supports very well you high elevation hypothesis. The authors could even use these lapse rates to document the paleo-elevation.

Figure 11: This figure is really good!

Brice
* * *

---

## Author Comment (AC1) · 28 Aug 2020

"Muñoz-López and co-authors document evolving fluid sources and conditions during multiple stages of deformation through a combination of field, petrographic, and geochemical analysis of an exhumed thrust in the Pyrenees. The manuscript is generally well written, with only minor grammatical or stylistic edits, and the quality of the work is robust. I do think that the manuscript could be improved with relatively minor edits, specifically with regards to motivation, descriptions of geologic units, and inclusion of a better synthesis figure".

[Figure]

We thank the referee for his constructive comments, which greatly helped to improve the quality of the manuscript.

General Comments:

"The motivation for the study should be stated clearly and much earlier in the document. The clearest iteration that I found was on page 13, the first sentence of section 5.4. It would be helpful for this type of statement to appear in the abstract, and in the introduction".

The main objectives of this contribution were already stated in the text. However, clearer sentences explaining the purpose and objectives of this paper are now included in the abstract and the introduction section.

"For all the discussion of basement, the age, lithology, and metamorphic grade were only briefly discussed, and relatively late in the text. It would be helpful to expand and highlight the section on the basement geology, and specifically previously published geochemistry, instead of the broader regional tectonism. It is particularly confusing because you repeatedly refer to "crystalline basement" but we learn that the basement is composed of Paleozoic slates, phyllites, sandstone, mudstone, limestone, conglomerate, and shale in Figure 11 and at the very end of the background geology, and if these are not the basement rocks then I'm completely lost. This is perhaps relevant because the Sr data is really more about interaction with older rocks with a different radiogenic signature, not rocks that are "crystalline" or not, correct? "

We expanded the description of basement rocks in the geological setting describing the age, lithology, metamorphic grade and involved tectonic event.

We agree with the reviewer that the Sr data from the basement is likely related to the interaction between vein-forming fluids and older rocks with a higher radiogenic signature. Therefore, we removed the phrase "crystalline rocks" as it was not properly used. Instead, we now refer to "basement rocks".
"I would like to see a figure of chemistry/temperature/source and deformation/tectonic events through time, rather than the 3D block diagram. The source and direction of infiltration and upwelling are rather speculative, but putting your observations into a more linear, temporal framework would be really helpful for me (see for instance my quick quick sketch). Similarly, annotating the figures showing geochemistry with some additional information about the inferred timing of cements, or associated structures, would be very helpful."

We combined the final 3D diagram with a temporal framework summarizing the tectonic and geochemical evolution of the area through time, similar to the sketch suggested by the reviewer.

All figures showing geochemistry have new information about the structure related to each calcite cement. We also added arrows to better observe the relative timing of all cements in the graphs. Therefore, with this new information it is easier to observe the geochemical evolution over time.

"What you meant by "timing" was a little misleading. At first, I assumed you were talking about relative timing from cross cutting relationships, but then you mention radiogenic ages, but then you acknowledge that you did not get ages. . . I think it would be better to be more specific, and upfront, about these being relative ages that are broadly linked to specific styles of tectonism".

We firstly stablished the chronological order of the observed structures based on cross-cutting relationships of (micro)structures. Then, we attempted to validate our interpretations by means of U/Pb geochronology. However, as we did not obtain absolute ages from U/Pb data, we use relative ages instead of absolute ages for discussion. To better clarify this, we explained in the results section that we did not get absolute ages and therefore, we are discussing with relative ages. In addition, we now use the term "relative timing" throughout the manuscript.

"This was originally a specific comment, but I think it applies to the whole manuscript:

Many hydrothermal systems are ultimately derived from meteoric fluids, and I agree that there is a clear and reasonable distinction between your cooler, younger fluids and hotter fluids, but a short sentence relating the importance of fluid-rock reaction and chemical evolution at higher temperatures and longer times would be helpful, because I think you are really making the point that you can track fluids because of that residence time at depth, not their ultimate source".

We added a paragraph at the end of the discussion section 5.3 in which we summarize the geochemical evolution of cements Cc1a to Cc5 and its implication with water-rock interactions and with changes in the fluid regime (upward vs. downward fluid migration). This geochemical evolution through time is clearer in the new final sketch (Fig. 13).

Specific Comments:

"Line 13: "timing" is a bit vague here. Because absolute age in faults and fractures is such a hot topic, I think it is important to specifically describe relative ages throughout."

Changed. We have specified that we refer to relative timing of fluid migration and vein formation.

"Lines 17-21: Crystalline basement is vague, especially in light of the abundance of low grade and unmetamorphosed rocks you show in later figures. Also, the distinction between deeply circulated meteoric and hydrothermal fluids... Is this just a matter of temperature or chemical evolution?"

We removed "crystalline basement rocks" as it was confusing and we now describe only "basement rocks" or "basement lithologies". The distinction between deeply circulated meteoric and hydrothermal fluids is based on both, chemical evolution and temperature. For instance, the clearest evidence of the presence of meteoric fluids (which precipitated cements Cc1a and Cc2) comes from the $\delta18O_{fluid}$ (yielding typical meteoric values between -6.4 and -0.3‰). By contrast, the clearest evidence of

the presence of hydrothermal fluids (which precipitated Cc3 and Cc4) comes from the relative high temperatures (up to 208°C) in comparison to the burial depths (probably less than 1 km). In both cases, the high 87Sr/86Sr ratios point to deep circulation and interaction with basement lithologies.

"Line 33: I'm not sure how understanding past fluid flow helps use understand the current configuration of a mountain belt. Perhaps it informs the factors leading to the current configuration? Additionally, I think there is a missing comma after "through time"

We agree with the reviewer and the paragraph has been changed accordingly. The missing comma has been added.

"Line 51: You say only a few, but cite 7 papers working on similar topics. Also, "On the other hand" does not seem necessary here."

Seven papers work in similar topics in the Pyrenean basement, but all are focused on the same thrust system. This has been specified in the text.

"On the other hand" has been removed.

"Line 100: These are the crystalline basement rocks? I think the geologic background should be more clear about rock types much earlier."

We agree with the reviewer, "crystalline rocks" was incorrectly used and we now use "basement rocks" instead. Also, as said in a previous comment, we expanded the geology of the basement rocks and deleted the word "crystalline" in order to avoid misunderstandings.

"Line 105: I'd like to see a short description of why samples were selected before launching into all the analysis, i.e. the field methods component, which is well documented in your figures. You include a lot of structural context in "Results" but it is difficult to evaluate whether 35, 12, or 8 samples is enough without some description of the structures that are present."

We firstly stablished the different vein generations observed in all fracture sets and fault-related deformation. Then, we selected representative samples of all these vein generations and related host rocks. This is now explained in the text.

"Line 121: Extra comma in "then, they"

The extra comma has been removed.

"Line 138: Extra comma in "one, keeping"

The extra comma has been removed.

"Line 162: First mention of U-Pb geochronology. So is this about absolute age?"

As said before, we did not get U-Pb data and therefore, our interpretations are based on relative timing. This has been specified in the new manuscript.

"Line 191: I understand the shorthand for regional foliation being Sr, but it is unfortunate that this paper also discusses strontium. Perhaps use Sr?"

We do agree with the reviewer and we have changed the shorthand for regional foliation (we now use S1 instead of Sr). Also, in order to simplify, we have changed the shorthand for thrust zone foliation affecting the hanging wall and footwall (we now use S2 instead of SD and SSP).

"Line 241: Perhaps "steeply" dipping?"

Exactly, it has been changed.

"Line 246: "frequently affect"? Commonly may be a better choice; frequent implies some element of time."

Changed. We now use commonly instead of frequently.

"Lines 225-253: It may be imbedded in the figures, or I may have been tired at this point, but I felt like the cross cutting relations or structural context for sequencing was a bit week with veins and cements 3-5."

Veins V3 to V5 are located in different structural positions (e.g., Fig. 3). Therefore, we could not observe crosscutting relationships between them in the field. However, these veins postdate the thrust-related structures and their formation is compatible with the Neogene extension (as explained in the discussion section). In the case of veins V3 and V4, their related calcite cements Cc3 and Cc4, have a similar geochemistry (Fig. 9 – 12). This supports precipitation of these cements during the same tectonic phase and associated with the same fluid flow event (i.e., although they precipitated in different structures, they are probably contemporaneous). Veins V5, as specified in the text, probably precipitated during the latest stages of extension, when the fluid regime changed from upward fluid migration to downward percolation of fluids, similar to models proposed in similar settings (e.g., Cantarero et al., 2014b).

"Lines 262-266: This could go in a table."

This information is already on table 1. In this part of the text, we describe the isotopic results and the observed isotopic tendencies.

"Line 301: Oh, there is NOT absolute geochronology. This passage should appear much earlier."

This passage has been moved to the results section. There, we also explain that we are dealing with relative timing and not with absolute timing.

"Line 304: An example of the basement lithologies and ages being clearly and simply defined."

As suggested by the reviewer in another comment, we added a similar description in the geological setting in order to better describe the basement geology, linking lithologies, ages and tectonic events.

"Line 330. Capitalize "Calcite..."

Done.

"Lines 355-361: Emphasis on channelized is a little confusing, because you also say that cements in different structural positions precipitated from the same fluids, which then begs the questions How wide are the fluid flow channels? Do you have samples from inside and outside of these channels? I think this invites a lot of extra scrutiny, and as it is written it is too vague."

Preferential fluid circulation along the thrust is evidenced by the exclusive presence of calcite cements (Cc1a and Cc2) within the thrust zone. As these cements (Cc1a and Cc2) precipitated during the same tectonic event but in different structural positions within the thrust zone, they likely precipitated from the same fluids, progressively increasing the fluid-rock interaction from the thrust plane (Cc2) towards the hanging wall (Cc1a). We have explained this in the text and the word "channelized" has been removed in order to avoid confusion.

"Lines 385-390: I think you do a good job defending your interpretation of fluid sources here."

We are glad to hear that from the reviewer.

"Line 393: replace "than" with "as"?

Replaced.

"Line 402: Probably the clearest statement of purpose in the manuscript."

We added a similar statement of purpose in the abstract and in the introduction. In the 5.4 subsection, we specified that we are assessing the influence of basement rocks on fluid chemistry.

"Line 464: "Common reservoir" implies a system in hydrologic or pressure communication. I don't think this is necessarily supported. Rather, you could claim that fluids are sourced or resided in similar basement rocks."

We agree! In the new manuscript we state that these fluids are sourced and/or interacted with similar basement rocks.

"Lines 469-470: Great Basin and Basin and Range Province are not necessarily synonymous, but they are in this case and therefore redundant."

Changed. We now use only Great Basin in the text and the two references have been merged.

"Line 495: Although you do include a caveat, I think the phrase "long-term" implies persistence, which is not necessarily true. I think places in the text that you describe long term fluid flow should be revisited."

We removed the term "long-term" as we have no evidence of the persistence of fluid flow. Instead, we state that hydrothermal fluids migrated in Neogene times and in present times and that the circulation of these fluids could be continuous through time or related to different pulses of fluid flow.

"Line 790 (Table 1): alignment issue after sample C6.II in some columns impacts readability. Also check on reason for differences in significant digits for reported values of the same quantity (for instance, see d18O and d13C columns)."

The alignment issue has been corrected and the significant digits have been revised.

"Line 800 (Table 2): check grammar "when is darker". A scale bar for color might be useful so we know what is considered high or low or how divisions were made (are they quartiles, or relative to some standard?)."

The grammar has been checked. In table 2, for each element, the darkest green points qualitatively to the highest concentration and vice versa. This has been explained in the text and a scale bar has been provided.

"Line 805 (Figure 1): Including generic rock types instead of or in addition to ages would be helpful, see generic comment about describing the lithology of the basement rocks."

We added generic rock types in addition to ages of the map performed in the study area.

"Line 855 (Figure 9): The relative timing of cements can be inferred from their order, but perhaps arrows on the graph showing fluid evolution over time, or some other way to relate these to types of structures or tectonic events would be useful. See general comment about a timeline figure."

Arrows have been drawn in the graph to show the geochemical evolution of calcite cements over time. In addition, the structure related to each calcite cement has also been provided.

"Line 865 (Figure 10): Again, this could be combined with a figure showing fluid evolution in the context of other events or structures, rather than leaving it to the reader to relate samples and setting. Add more text on the graphs to help guide me."

A graph showing the structure related to each calcite cement and their evolution over time has also been provided.

"Line 874 (Figure 11). Shales and mudstones don't seem particularly "crystalline", but you refer to crystalline basement a few times in the manuscript. I can understand why dilatant crystalline rocks may be geothermal reservoirs, but why these low grade rocks? Is it just that these rocks exist at deeper, hotter depths? Brown line seems a bit high (it cuts off a few basement values), how it was chosen?"

As previously stated, the term "crystalline rocks" has been removed, we refer now to "basement rocks".

The hydrothermal character of the fluids is inferred from their high temperatures (up to 208°C), which are higher than values expected by normal geothermal gradients (if we consider < 1km burial depths, according to Saura, 2004). Therefore, the involved fluids probably warmed at greater depths and then migrated upwards through Neogene faults, flowing fast enough to maintain their high temperatures or at least, to be in

thermal disequilibrium with the surrounding rocks. This has been better explained in the new manuscript.

We selected the limit between basement and cover values taking into account both the previous contributions and our own results. As shown in Fig. 11, all calcite cements precipitated from fluids that have circulated through cover rocks have 87Sr/86Sr ratios clearly lower than 0.710. By contrasts, all calcite cements precipitated from fluids that have circulated through basement rocks have 87Sr/86Sr ratios greater than 0.710 (i.e., these calcites reflect a higher radiogenic signature, similar to that reported for basement rocks). The only exception is observed in reference 14 and reference 16 (Fig. 11). Reference 14 is located in Plan de Larri, at the transition between basement and cover structures. In this location, the lowest 87Sr/86Sr values (<0.710) are found in relatively undeformed Cretaceous carbonates, which are lithologies widely exposed in the Pyrenean cover (in the Southern Pyrenees). This explains the little overlap between these lowest values and those of the sedimentary cover. On the other hand, as explained in the manuscript, authors from reference 14 (McCaig et al.,1995) used the same limit (i.e., 87Sr/86Sr = 0.710) to differentiate between values derived from the unaltered limestone protolith and the thrust-related carbonate mylonite affected by a fluid carrying radiogenic Sr. Values of reference 16 belongs to low grade metamorphic rocks. Although the lowest values of this reference overlaps those of the sedimentary cover, no information about synkinematic veins have been provided, which is the focus of the comparison in this contribution.

"Line 885 (Figure 12). Again, I'd rather follow the changes over time, so instead of looking at changes in Mg, then Fe, then Mn. . . plot these values in the context of other events and features and then we could see what was happening with Mg when Sr goes up, for instance."

We have modified this figure according to the reviewer comment. Now it is easier to follow changes in the elemental composition over time.

"Line 890 (Figure 13). This is a fine figure, but I don't know that it adds a great deal to the story, other than showing what you have already described fairly well in the text. If you do keep it, I think showing warmer colors for hotter fluids may be more intuitive."

As said before, we added a sketch summarizing both the tectonic and geochemical evolution of the studied area. We also changed colors of the arrows that indicate fluid migration, that is, we now use red and orange colors for warm fluids and green colors for cold fluids.

---

## Author Comment (AC2) · 28 Aug 2020

Review of the paper "Influence of basement rocks on fluid evolution during multiphase deformation: the example of the Estamariu thrust in the Pyrenean Axial Zone" by Daniel Muñoz-López, Gemma Alías, David Cruset, Irene Cantarero, Cédric M. Jonh, Anna Travé

"This paper by Munoz-Lopez et al. reports structural and geochemical evidence of several multi-phase fluid-flow along the Estamariu thrust located in the Pyrenean Axial

[Figure]

Zone combining detailed structural and microstructural observation to O, C, Sr and $\Delta 47$ isotopes. The techniques employed here are adequate and I'm glad to see application of $\Delta 47$ thermometry in the Axial zone. The dataset is sound, and the conclusions are reasonable. The authors document a complex fluid-flow history along the thrust during Alpine compression and Neogene extension involving different sources of fluid at different temperatures. This paper should be published as it is an important regional contribution. As stated by the authors, there is a long list of works studying the fluid-flow along thrust faults affecting the sedimentary cover (e.g. Southern Pyrenean zone), but just a few focuses on the basement from the Axial Zone."

We thank the referee for his positive comments and detailed reviews.

"Although this work should be published, there are several points that need to be addressed/commented during revision.

1. The structural analysis and description are very detailed but somehow confusing. Here are some suggestions that could clarify the description:

- Adding sub-titles such as 4.1.1. Adding sub-title such as Hanging wall, Main Thrust, and Footwall would help a lot."

Sub-titles have been added to clarify the text. (i.e., 4.1.1. Hanging wall; 4.1.2. Thrust zone and 4.1.3. Footwall).

- "I would also recommend adding a general schema synthetizing all the relationship between the observed structures and microstructures. The current Figure 3 does that, but it is still confusing."

As pointed by the reviewer, Fig. 3 represents a general sketch that summarizes the relationships between the observed structures. We have modified this figure in order to avoid confusion. For this, we simplified the terminology used to describe the different structures (see for instance the answer to the following comment) and added arrows to match such structures with their correspondent stereoplots.

- "Also, I found the adopted typologies for structures "Sr, Sm, Ssp, SD, etc. . ." confusing and did not catch up what the subscript letter ('r' and 'm') refer to. What about calling these foliations the same way (unless they are associated to different tectonic phases) and just describe them (morphologies, intensity, orientation) in the sub-section. This would simplify understanding of the numerous stereoplots presented in Figure 3. I also noticed that some of these foliation typologies are not called in the text. For example in line 199: "The foliation within the thrust zone affecting the Devonian hanging wall strikes NW-SE and dips 40 – 50o NE, similar to the regional foliation in the protolith, but it is more closely spaced, generally between 0.2 and 1 cm (Fig. 6A, B)." Specify in the text if this foliation corresponds to "Sr". The same comment is applicable for all the section 4.1."

We changed and simplified the shorthand used to describe the observed structures. For regional foliation we now use S1 (instead of Sr). The foliation associated with the thrust (i.e., the thrust zone foliation) is now called the same way (S2), even if it affects the thrust zone deforming the hanging wall or the footwall (we use S2 instead of SD or SSP). We do not use shorthand for layering magmatic anymore (i.e., we removed the shorthand Sm) because layering magmatic is an inherited fluidal structure that is not discussed in detail.

All foliation typologies are clearer and better described in section 4.1.

- "You mentioned pressure solution surfaces e1 and e2 but it was not clear on which basis they were differentiated. Is there any cross-cutting feature? The orientation of these features (if any) are not presented on stereoplots."

These stylolites were only observed locally at microscopic scale and therefore, their orientations are given according to more relevant structures (i.e, they trend subparallel to the thrust zone foliation). This is stated in the text.

We could not find crosscutting relationships between these structures, however, we differentiated them according to their spacing and crosscutting relations with veins V1a.

Stylolites e1 are less spaced and are always crosscut by veins V1a, whereas, stylolites e2 are more abundant and postdate V1a (as they developed as sutured areas between the host rock and veins V1a). This has been clarified in the new text.

2. "The authors report warm temperature fluid-flow event (up to 200°C), presumably hydrothermal fluid for the CC3 and CC4 calcite phases. As you know, $\Delta 47$ composition of carbonate may be altered by $\Delta 47$-reordering when carbonate experienced temperature in excess of 200°C (maybe lower temperature). Although, I'm convinced that these hydrothermal events did not altered previous carbonate phases (cc1, cc2), I would like to see the authors discussing potential (or not) $\Delta 47$-reordering and how it could be ruled out. Here are some ways to discuss that: - Is there some metamorphic/Fluid Inclusion/chronology work in the same area reporting temperature-time relationship of this hydrothermal event? If this hydrothermal event if short, the solid-state $\Delta 47$ probably did not occur; - Is the thermal history of the area constrained by other studies? If it is the case, the authors could use re-ordering models (e.g. Stolper and Eiler 2015; Lloyd et al., 2017) to see if the $\Delta 47$ composition of the different calcite phases may experience re-ordering. - Alternatively, the authors may acknowledge that further re-ordering is possible but unlikely due to the short time."

We agree that clumped isotopes may potentially be altered due to the relatively high temperatures. However, as suggested by the reviewer, we also consider that this is unlikely because of the relative short time of hydrothermal fluid migration in the study area and because there is no evidence of calcite recrystallization and/or solid-state reactions. We have explained this in the new manuscript, in section 5.3 of discussion.

3. "I have noticed few poorly constructed sentences. I would suggest the English to be checked before re-submission. I won't make any comment on that as I'm myself always struggling with English."

The English grammar has been revised and improved.

Minor comments:

"l. 28: Deformation associated with crustal shortening is mainly accommodated by thrust faulting and related fault zone structures: Add references."

References added (Mouthereau et al., 2014; Muñoz, 1992a; Sibson, 1994).

"l. 30: "The reactivation of faults may produce changes in the hydraulic....": Add References."

References added (Arndt et al., 2014; Barker and Cox, 2011; Cantarero et al., 2018; Cruset et al., 2018a; Lacroix et al., 2018; Travé et al., 2007a).

"l. 174: Which fractionation curve is used to calculate the oxygen isotope composition of water?"

The fractionation equation used to calculate the oxygen isotope composition of water is the one from Friedman and O'Neil (1977). This is stated in the new text.

"l.180 – 181: "The main slip plane is undulose, producing changes in the strike direction and dip, and generates a 2 – 3 m thick thrust zone, which is thicker in the hanging wall, up to 2.5 m thick": Do you mean the thrust fault consists to a deformation zone affecting both hanging wall and footwall, with deformation zone thicker in the HW?"

Exactly! The thrust zone affects both the footwall and hanging wall. This has been specified in the new text.

"l.187: "In the studied outcrops, the Devonian Rueda Formation from the hanging wall is characterized by a well-bedded alternation of dark to light grey limestones with dark grey shales": Does this refer to S0, Sr, Sd? Please specify."

It refers to S0 (bedding). It has been better explained in the text.

"l.190-192: "Deformation in the Devonian protolith (i.e., outside the thrust zone) corresponds to a decametric anticline (Fig. 2B), which has associated an axial plane pervasive regional foliation (Sr) concentrated in the pelitic intervals (Fig. 5B)": looking at the stereoplot from Fig. 3, the bedding (S0) seems to define a fold oriented E-W

(although only based on 3 measurements). In contrast the Sr does not seem parallel to the axial plane and is more or less oriented N-S (even slightly folded). How can you explain this?"

As observed in Fig. 2, the fold in the hanging wall corresponds to a tight and SW verging anticline. Looking at the stereoplot (Devonian protolith), the observed planes define the fold limbs, which are approximately oriented WNW-ESE and NE-SW. The intersection between these planes (through the bisector angle) defines the axial surface, which is oriented ∼NNW-SSE, similar to the regional foliation. This has been explained in the new text.

"l. 203-204: "When present, these stylolites are very systematic with densities between 5 and 8 stylolites/cm." Should the intensity be given in number/cm2?"

This is a rough estimation because stylolites e1 are only locally observed at microscopic scale. Therefore, in the new manuscript, we give the stylolites intensity by means of their spacing, this is, they are 1 – 2 mm spaced (Fig. 6C).

"l. 201 – 202: "At mesoscale, SD has related shear surfaces (Ci) defining centimetric S-C-type structures, again indicating reverse kinematics (Fig. 6A).": Do you have a closer view of the C-S relationship?"

We improved the quality of the picture showing S-C structures. The S-C relationships is better seen now.

"l.220-222: "The vein cement (Cc2) is milky white in hand sample and consists of up to 3 mm blocky to elongated blocky crystals (Fig. 6G) with a dull to bright orange luminescence (Fig. 6H).": To me, e2 and V1b are clearly cogenetic as their crosscutting relationship are ambiguous (as stated by the authors). It is also interesting to see that V1b is extensional (Mode I) but also show mode II with conjugate opening (Fig. 6F). In any case all these structures formed under the same field stress and can be assumed contemporaneous."

We totally agree with the reviewer and, as it is explained in the text, these observations indicate that these structures developed coevally.

"l.226: "They are parallel or locally branch off cutting the foliation planes in the subsidiary thrust zone": What are the textures of these veins? They seem to show interdigitated texture in agreement with extension opening. These are important as they give indication of the opening regime and stress field."

As explained in the text, although veins V3 locally branch off, they are mainly parallel to the foliation planes and therefore, we could not observe geometrical features (in the field) indicative of the stress regime. However, the fibrous texture of the calcite crystals, growing perpendicular to the vein walls and to the foliation planes, indicates their extensional character. This extensional opening postdates the thrust zone foliation and is compatible with the Neogene extension. On the other hand, the vein cement (Cc3) has a similar geochemical composition to the cement present in the Neogene normal faults (Cc4). This observation seems to indicate precipitation during the same tectonic event and associated with the same fluid regime. This has been stated in the new text.

"l.246: You state here v4 for fault. However, you previously used V labels for veins. It is confusing even if we expect slickenside onto these faults."

We use V labels (V1 to V5) as a shorthand for calcite veins instead of the related structure type (opening fracture or fault). This has been clarified in the new text.

"l.252-253: "Shear fractures (V5) are locally mineralized with a greyish microsparite calcite cement (Cc5).": Any Figure to document?"

We added two images of these veins, a field image of the vein and a microphotograph of the vein-related cement.

"l. 400: Huyghe et al. (2018): "Impact of topography, climate and moisture sources on isotopic composition ($\delta$18O & $\delta$D) of rivers in the Pyrenees: Implications for topographic

reconstructions in small orogens" reported new isotope lapse rates for the Pyrenees. This study should be cited here as it supports very well you high elevation hypothesis. The authors could even use these lapse rates to document the paleo-elevation."

This reference has been added because, as suggested by the reviewer, this study supports our interpretations. The paleo-elevation have not been calculated because according to Huyghe et al., 2018 the relationship between the isotopic composition and the elevation is not clear in the Southern Pyrenees. According to these authors, the elevation does not seem to be the only parameter controlling the isotopic composition in this part of the belt.

"Figure 11: This figure is really good!"

We are glad that the reviewer likes this figure.

---

## Author Response (AR2)

Revised Submission

**Topical Editor Decision: Publish subject to minor revisions (review by editor)** (29 Sep 2020) by
Randolph Williams

Comments to the Author:

Dr Dear Muñoz-López,

**Thank you for submitting a revised version of your manuscript that addresses reviewer comments. I feel that significant progress has been made when compared to the original submission. A bit more work, however, will be required in order for the submission to be an acceptable state for publication in Solid Earth. I would like to reconsider this manuscript following minor revisions. Please see below for details.**
**The majority of my current concerns for this manuscript are related to presentation. There are a few sections of text that are still a bit rough around the edges, sometimes in areas where clarity is particularly paramount to understanding the presented work. I have made many specific comments below, but will reiterate some of the main issues with the presentation / writing quality here.**

Dear Dr Randolph Williams, thanks very much for your constructive comments. In the new reviewed manuscript, we have addressed all suggestions. We agree with you that the manuscript has been highly improved after reviewer comments.

**1) The writing has a tendency to confuse fact and interpretation, mainly in the discussion. As an example, consider the text: "As veins V1a and V2 developed during the Alpine reactivation of the Estamariu thrust, the geochemistry of their related calcite cements Cc1a and Cc2 record the fluid system during this tectonic event." It is only your interpretation that the veins formed during Alpine reactivation. The structural data support it, but without actual geochronology this cannot be known independently. The language should reflect that.**

As proposed by the editor, we have modified the text to differentiate between facts and interpretations. For this, we have added sentences like "these results indicate that…" (as has been suggested by the editor in other comments).

**2) The age descriptions of various events need to be revised. Right now there is a tendency mix global (i.e. Neogene) and local (i.e. Variscan) terms. The readership for this journal is international, and those outside of Europe may not immediately know the age range of the latter.**

We now provide the age of the tectonic events discussed in this study, this is, Late Paleozoic for the Variscan orogeny and Late Cretaceous to Oligocene for the Alpine compression.

**3) The overall text is, in my opinion, considerably longer than it needs to be. Significant shortening could be accomplished in most sections. Methods that produced no data (i.e. U-Pb) need not be mentioned. This is also true of aspects of the geological setting description that do not directly impact the overall interpretations presented here (whether or not this is the case I will leave to you).**

We have shortened several sections. In the geological setting, we have removed data that is not relevant for this study. The methods have been moved to supplementary data and we only describe the more relevant information in the manuscript. The discussions and conclusions have also been shortened for brevity and clarity (see the following comment).

**4) The overall quality of the writing tends to decrease toward the end of the manuscript. Please revise the manuscript for brevity and clarity. In the discussion, for example, the last 4 sections tend to either reiterate the same points continuously, or call upon complicated theoretical considerations to underpin interpretations that you have actual data to support (see comments related to section 5.5). These sections would quite easily be condensed into a single section just a few paragraphs in length.**

The last four sections in the discussion aim to explain different points. For instance, in sections 5.2 and 5.3 we explain the fluid system during the Alpine (Late Cretaceous-Oligocene) and Neogene times, respectively. For this, we use all our geochemical analyses performed in the study area.

In section 5.4, we compare the $^{87}Sr/^{86}Sr$ ratios of the studied calcites with those obtained in other structures from the Pyrenees. This comparison allows us to provide a specific limit values between fluids circulating through the basement and fluids circulating through the cover.

In section 5.5, we highlight that the fluid system in the study area is similar to the fluid system reported in the Catalan Coastal Ranges (CCR), suggesting a common fluid behavior in both places (this is an interesting regional point of view, in the NE part of Iberia). In this section (5.5) we are not only comparing the $^{87}Sr/^{86}Sr$ ratios, but also the temperature and the hydrothermal character of the fluids circulating in Neogene times and in the present.

All these sections have been shortened for brevity and clarity, removing information already said in other sections.

**From a scientific perspective I think things are largely in good shape, but please do address the issue of negative clumped isotope temperatures and their impact on d18O-fluid interpretations.**

We agree with the editor that negative clumped temperatures are rare. Samples from veins V5 were too small and therefore, we only had two replicates and the reproducibility was rather poor. One replicate is at 0.707 and the other one at 0.767. Therefore, as this could compromise the calculated T and the $\delta^{18}O_{fluid}$, we decided to remove this data from the manuscript. We believe that our main conclusions do not change without this data.

**Other comments**

**19: I am skeptical that clumped isotope analysis allows us to know temperature this precisely even from an analytic perspective, let alone a natural one. Suggest "50 - 100 C", here and throughout.**

We now give the range of temperatures suggested by the editor in the whole manuscript.

**21: Suggest "...hydrothermal fluids that had interacted with crystalline basement rocks and been expelled…"**

We agree with the editor. We have changed this paragraph accordingly.

**23: "Cc5 cements"**

Changed.

**A general comment I have on the abstract is that it is stating interpretations as a matter of fact. Please rephrase to make clear that these are your interpretations, citing the relevant observations / data where necessary to support your case.**

We have rephrased the abstract to differentiate interpretations and facts. We hope that the abstract is clearer now.

**46-47: Those unfamiliar with the tectonics of the area are unlikely to know the age of the events you are referring to here.**

We have provided the age of the tectonic events that we are referring to.

**49-50: It is not clear to me why a lack of knowledge of the age of a fault precludes examination of fluid-fault interaction?**

Rocks from the Axial Pyrenees have been affected by successive deformation events and the observed fault systems may have been formed and/or reactivated during different episodes. It is not always clear what structures and veins belong to a specific event of faulting (due to the transposition of different deformation phases, usually with lack of crosscutting relationships between them).

**53: Suggest, "In contrast, few studies have examined the relationship between deformation and fluid flow in the Paleozoic basement…", or similar.**

This sentence has been changed according to the editor suggestion.

**66: Please do not mix local age/orogen terms with established geological periods, which are global. The latter are preferred.**

The age of the tectonic events has been added.

**81-82: Same comment as above.**

This paragraph has been removed from the geological setting as it provided additional information that was not need it for the specific focus of the manuscript on the Estamariu thrust (as suggested by the editor in another comment).

**105: "Basically"?**

In this sentence, we are describing the main Upper Ordovician lithologies. We have change the word "basically" for "mainly".

**General comments on geologic setting: I would encourage you to consider shortening this section. It is not clear to be that all of this information is required given the specific focus on the Estamariu thrust.**

The geological setting has been shortened and we only provide relevant information to understand the study area.

**114: Please restructure this sentence. Too many "ands" and not enough commas.**

We have rewritten the sentence.

**115: "The structural data includes the orientation of bedding, foliations, and fractures in addition to cross-cutting relationships and kinematics".**

Changed accordingly.

**124: As it relates to isotope analysis, "isotopy" is not actually a word. Suggest "...selected for stable isotope analysis".**

We agree with the editor and have changed this sentence accordingly.

**General Comments on Methods: These can and should be shortened substantially. The "dirty details" of the analyses should be moved to a supplemental document, leaving the main text to focus on what was done at a higher level.**

The methods section has been shortened. The more relevant aspects are explained in the manuscript. Additional data and procedures have been moved to a supplementary data document.

**197-199: I thought it was strange that the U-Pb work was not mentioned until the methods. If this didn't produce anything useful for the present study then it would be best to remove it all together. I would note, however, that if the problem was low U and high Pb then you now have Pb isotope data, which previous work has shown to be very effective in tracing fluid origin / pathway information in faults (see Williams et al., 2019, Journal of Structural Geology - Radiogenic isotopes record a 'drop in a bucket'–A fingerprint of multi-kilometer-scale fluid pathways inferred to drive fault-valve behavior).**

As suggested by the editor, we have removed the paragraph explaining the U-Pb methodology because we did not get any result. According to the lab, the problem was probably related to the low U and

high Pb contents (i.e., concentration measured in ppm). As these values were not suitable for dating, the Pb isotopic analysis was not measured.

**202: "Subordinate"**

Changed.

**216: "Points to…"**

Changed.

**210-245: The orientation information here is very hard to follow. Figure 3 helps but more could be done. Suggest adding a stereonet figure that shows the average orientation of each of the named structural elements, such that their relationship to each other is easily visualized.**

The average orientation of the main structural elements discussed in this study has been provided in a new stereoplot (also in Fig. 3). Now it is easier to visualize the relationships between the main studied structures.

**288-293: The D47 value will not be interpretable to most. You could simplify this text by just stating that D47 values were converted to temperature using Kluge and then simply report the temperature range for each generation.**

We agree. We have changed this paragraph according to your suggestion.

**304-307: Similar to U-Pb comments, it is difficult to believe that Nd/Nd values for just two samples shed any real light on the story. This system is highly atypical in studies of fluid-rock interaction, and would likely be difficult to interpret confidently even if a lot of data were available. Seems like two data points provide little more than speculation.**

We agree with the editor that the Nd data is very limited. For these reasons, we useg this data only to support the interpretations that we already obtain with other geochemical methods. Previous studies have shown that Nd isotopes do not undergo mass fractionation during precipitation. Therefore, they may be good tracers of fluid-rock interactions (see for instance Voicu et al., 2000; Barker, 2007; Barker et al., 2009).

**318: Consistent age terminologies please.**

We now give consistent age terminologies (i.e., Late Paleozoic to Neogene).

**340: I do not see how fibrous crystals orthogonal to fracture walls indicate extensional deformation? I have seen identical structures in a variety of thrust environments. Are you implying that syntaxial veins cannot form during thrust/reverse faulting, or is the "extension" in this case referring to the fracture only?**

We do not mean that syntaxial veins cannot form during thrust faulting; we are only referring to our particular study case.
Veins V3 are oriented parallel, but locally crosscut, the thrust zone foliation. This foliation is consistent with compressional deformation (which is oriented ~perpendicular to the foliation surfaces). In contrast, fibrous crystals in veins V3 indicate opening (extension) in a direction perpendicular to the fracture walls and therefore, perpendicular to foliation surfaces, which is not compatible with the mentioned compression. For this reason, we conclude that these veins formed along previously developed foliation surfaces, postdating the thrust activity.

**353-354: This is an interpretation that has the effect of being stated as a fact here. Please be careful with phrasing related to this here and throughout the manuscript.**

We agree and we have rewritten the text.

**370-375: Please rephrase these lines for clarity. A big part of your argument rests here, but the text related to it is a bit cumbersome.**

We have divided this paragraph into smaller sentences to clarify the text.

**390-393: Interesting. Fluids at ~200 C at a depth of only a few hundred meters should have experienced boiling. Any evidence that would suggest that happened?**

The obtained temperatures are certainly high with respect to reported burial depths in the area. However, we have no evidence of boiling or similar processes associated with these fluids.

**405-406: I do not see how any of your observations can reasonably support this assertion. I would argue we have no idea how long that event was, nor even how long or short it would have to be to result in the resetting of clumped isotope data. The line below is specific to temperatures in excess of 120 C, but the process of reordering is almost certainly kinetically controlled and may therefore scale exponentially with temperature. The only observation that can reasonably support the integrity of the other earlier cement generations is the lack of apparent recrystallization structures.**

We agree with the editor that the lack of recrystallization structures may support the integrity of the clumped isotope values of the earlier cements (this was already stated in the text). Therefore, we highlight this observation to validate our clumped isotope results. The other observation has been removed for brevity, as we have no evidence of how long the hydrothermal event was.

**408: "Crumpled" isotopes. Obviously a typo, but quite funny nevertheless.**

Exactly! It was a typo and it has been corrected. However, we agree that maybe one day we will be able to do crumped isotopes on carbonates!

**411-412: This is a bit of a sticking point for me. The stable isotope composition of calcite cannot indicate any type of fluid origin on its own. Only the calculated d18O-fluid can provide that information.**

We agree. However, our values fall within the range of freshwater carbonates (Veizer, 1992). We have better explained this in the text.

**416: How can the abundance or size of the veins be indicative of meteoric fluids? I have seen fault systems with a significant abundance of very large veins that formed almost exclusively from meteoric fluids.**

We agree that the abundance and size of the veins is not indicative of the fluid origin. We have removed this sentence. The meteoric origin of the fluids is indicated by the geochemical data.

**418: I'm sorry, but -5 C? That is ice for fresh water (which you seem to be inferring), even at high altitude. This suggests that perhaps your D47 data for this generation (at least) are skewing a bit lower than reality, which has implications for the d18O-fluid values you calculate. This certainly warrants some discussion.**

As explained above, we removed this data from the manuscript.

**440-458: I do not quite understand the point of this section given the last paragraph of the previous. Is it just to say that fluids circulating through mesozoic and cenozoic cover sediments have a different Sr isotope composition then those circulating through basement rocks? This is more or less what would be expected. I would also point out that, since you are referencing other studies that employed a 0.710, "limit value" for inferring interaction with older crystalline materials, that Williams et al. (2015, Geology) employed the same value for fluids in the Rio Grande rift, USA.**

The last paragraph of the section 5.3 was used to explain the geochemical evolution of the fluids and its implications on the fluid regime during the tectonic evolution of the study area. In contrast, in section 5.4 we compare the $^{87}Sr/^{86}Sr$ ratios of the studied calcites with those obtained in other structures from the Pyrenees. We specified this in the title of section 5.4.

Certainly, we would expect different $^{87}Sr/^{86}Sr$ ratios in basement and cover rocks. However, the interesting point of this section is to provide a limit value between these two domains.

We included the reference of Williams et al., 2015, which also attributed similar values to fluids interacting with basement rocks.

**497: Was all of the above discussion in this section meant to argue that the fluids precipitating the calcites examined in this study interacted with basement rocks? Surely your actual data (mainly Sr/Sr) demonstrated that fairly conclusively already? I am struggling to understand the utility of this section.**

As explained in the text, the aim of section 5.5 is to provide insights into the origin and evolution of fluids at regional scale. The Pyrenees and the Catalan Coastal Ranges (CCR) are two different ranges. We want to highlight that what we have concluded in the Pyrenees has also been reported in the CCR, what is interesting from a regional point of view. In this section (5.5) we are not only comparing the $^{87}Sr/^{86}Sr$ ratios, but also the temperature and the hydrothermal character of the fluids circulating in Neogene times and in the present.

**515-516: These types of sentences / ideas would be much better constructed as "Stable oxygen and clumped isotope analyses indicate that meteoric fluids were circulating through the fault during these periods, at temperatures of XX-YY".**

We now use these type of sentences in the conclusions.

**516-517: What does "increased the fluid-rock interaction" mean, and what is the evidence for it?**

We refer to the extent of fluid-rock interaction. Two cements (Cc1a and Cc2) precipitated from fluids circulating through the thrust zone. Cc2 precipitated in the thrust plane and Cc1a in the hanging wall. As Cc1a has $\delta^{13}C$ values and an elemental composition closer to the involved host rocks than Cc2, we suggest that the extent of fluid-rock interaction was higher in the hanging wall. This has been better explained in the text.

**523-525: This is a strange central conclusion to fall back on. It is exactly what would be predicted, and moreover it has been shown by several previous studies looking at Sr/Sr systematics in both fault cements and modern hot spring fluids. The conclusion feels a bit like it was put together as an afterthought, when I would argue it is one of the most important sections in the paper.**

We have reorganized the conclusions section in order to give more importance to the main findings of this study.

[revised manuscript text omitted]